# Assessing the value of integrating national longitudinal shopping data into respiratory disease forecasting models

Elizabeth Dolan [1,2] ✉, James Goulding [1], Harry Marshall [1], Gavin Smith [1], Gavin Long [1] & Laila J. Tata [3]

The COVID-19 pandemic led to unparalleled pressure on healthcare services. Improved healthcare planning in relation to diseases affecting the respiratory system has consequently become a key concern. We investigated the value of integrating sales of non-prescription medications commonly bought for managing respiratory symptoms, to improve forecasting of weekly registered deaths from respiratory disease at local levels across England, by using over 2 billion transactions logged by a UK high street retailer from March 2016 to March 2020. We report the results from the novel AI (Artificial Intelligence) explainability variable importance tool Model Class Reliance implemented on the PADRUS model (Prediction of Amount of Deaths by Respiratory disease Using Sales). PADRUS is a machine learning model optimised to predict registered deaths from respiratory disease in 314 local authority areas across England through the integration of shopping sales data and focused on purchases of non-prescription medications. We found strong evidence that models incorporating sales data significantly out-perform other models that solely use variables traditionally associated with respiratory disease (e.g. sociodemographics and weather data). Accuracy gains are highest (increases in $R^2$ (coefficient of determination) between 0.09 to 0.11) in periods of maximum risk to the general public. Results demonstrate the potential to utilise sales data to monitor population health with information at a high level of geographic granularity.

Between 2015 and 2019, respiratory disease (ICD 10 CODING (J00-J99)) was the underlying cause of 369,900 deaths across England and Wales[1]. Respiratory Disease, ICD 10 CODING (J00-J99), refers to infections and disease of the lungs and respiratory system including asthma, chronic obstructive pulmonary disease (COPD); bronchitis; emphysema, pneumonia and influenza. J00-J99 ICD coding does not include lung cancer, respiratory tuberculosis or cystic fibrosis. By 2020, and the arrival of the global pandemic, respiratory disease had become the leading cause of death[1], with COVID-19 listed on over 200,000 UK death certificates between February 2020 and July 2022[2].

With COVID-19 appearing to have established itself as a dominant and long-standing disease[2,3], and with public health services simultaneously facing increasing financial and logistical pressures, it has become more critical than ever to investigate better forecasting of Influenza-like Illnesses (ILI) and their likely impact on local and vulnerable populations.

### Behavioural data in integrated disease models
Response to COVID-19 also served to highlight the limitations faced by classical disease modelling in supporting prediction efforts. In

[1]N/LAB, Nottingham University Business School, University of Nottingham, Nottingham, UK. [2]Horizon Centre for Doctoral Training, University of Nottingham, Nottingham, UK. [3]Lifespan and Population Health, School of Medicine, University of Nottingham, Nottingham, UK. ✉ e-mail: elizabeth.dolan@nottingham.ac.uk

traditional forecasting models, individuals are assigned to just one of three states (susceptible-infected-recovered) and population behaviours are broadly assumed to be both homogeneous and static[4]. Evidence from the pandemic highlighted not only the key role micro-scale interactions between individuals play in disease transmission, but the highly dynamic and evolving nature of those interactions amidst social, economic and behavioural shifts within the population, especially at local levels[5]. In reaction to this, there have been renewed calls to directly integrate social and behavioural data into disease models[4–9], with the aim of producing more nuanced and adaptive forecasting mechanisms. Data that can provide insights into the evolving nature of population behaviours, from mobility patterns to nutrition and self-medication strategies, have been referred to as operational response data by Bedson et al.[4]. Although such event streams hold much promise, their use has remained limited due to lack of availability, access and appropriate levels of granularity. Consequently their practical utility is broadly untested.

While researchers have awaited development of the infrastructure to support use of digital footprint data within epidemiological models, they have continued to leverage more traditional data collection methods such as surveying and the harvesting of self-reports[4–8]. Surveying at scale, however, remains a logistically challenging endeavour, with sample sizes constrained by both budgets and capacity, along with the recognised potential for response bias[10,11], even when data is obtained directly via mobile applications[5,12]. When self-reports are obtained indirectly through analysis of social media accounts they too have similar response biases due to both memory failure and social desirability[13,14].

Alternative digital footprint datasets offer potential to provide a more sustainable and passive route to monitoring longitudinal health behaviours at scale. Such data can be used to supplement and complement qualitative insights, and act at a resolution that can reflect variations within local populations if key transparency, privacy and commercial sensitivity issues can be addressed. Privacy issues are particularly relevant when individual-level digital footprint data is accessed[15], yet aggregated digital data at local levels could still offer useful insights. While some datasets of this nature, such as mobility data from use of mobile devices, have already been applied to COVID-19 modelling with some success[16–18], no integrated disease models in the UK have incorporated sales data. Self-medication purchasing patterns hold key information about population-wide response to disease, yet their use in health surveillance remains scarce[19–21].

## Using shopping data logs to examine health

Recorded and stored by retailers across the UK, transactional shopping data consist of longitudinal, time-stamped purchasing logs, specified at store-level geographical granularity. One of the key advantages of such data-sets is that they are updated in real-time, offering capacity to investigate behavioural signals not only across a population, but over time. A variety of studies have reported the potential of such data to provide insights into population health[22–25]. Non-prescription medication purchase logs, in particular, have long been considered a valuable input into health surveillance systems[20,26,27], with Welliver et al. evidencing its potential to indicate influenza at community levels as far back as 1979[28]. Similarly, Hogan et al. showed a high correlation between respiratory and diarrheal outbreaks in children and sales of over-the-counter electrolyte products[29]. Socan et al. showed correlation between medicines for sore throat, antitussives, decongestants and mucolytics, and weekly influenza incidence; the sales of antitussives alone could predict increased incident rates[30].

There remains, however, contention as to the effectiveness of integrating such datasets in forecasting systems, a key issue given the surrounding privacy, ethical and transparency challenges of their usage[15,31]. Al-Tawfiq et al.'s review of health surveillance systems for emerging respiratory viruses reported only mixed results in the efficacy of monitoring of non-prescription drug purchases[32]; and while Pivette et al. systematic review of drug sales in outbreak detection demonstrated improvements in provision of earlier alerts, they also noted challenges in selecting indicator drug groups and poor-quality clinical surveillance data[19]. Davies and Finch's 2003 UK study over 3 winter periods (examining Nottingham City Hospital NHS, Boots and Reckitt Benckiser pharmacy data), reported that analysis of non-prescription cough/cold medications could provide a two-week warning of respiratory admissions peaks[33], yet Todd et al. found no significant correlation between retail sales of symptom remedies and cases at a sub-national scale for the 2009 UK influenza outbreak[34].

One of the challenges in assessing this prior research lies in the spatial and temporal granularity that researchers have had available to them. Disease transmission predominantly acts at local community levels - yet many studies have only been able to examine datasets aggregated to larger geographical regions. This has restricted many studies to broad-brush conclusions. Todd et al.[34] found that national non-prescription medication sales models, fitted using national spatial areas of the countries England, Scotland and Wales, and 10 health regions of England, were not useful for identifying weekly flu cases. However, models fitted using sub-regions defined by catchment areas for 156 English Primary Care Trusts showed some statistically significant positive correlations between flu cases and sales[34]. They concluded using retail sales at finer spatial scales may help augment existing surveillance and merit further study[34]. In this work we are able to address some of these prior limitations, investigating whether behavioural data (in the form of non-prescription medication sales) can increase the effectiveness of forecasting models for respiratory death at a weekly resolution, and acting at the far finer geospatial granularity of local authorities. This study investigates whether shopping sales data, drawn from over 2 billion transactions logged by a national UK high street retailer, at store locations distributed across all of England covering 314 Lower Tier Local Authorities (LTLAs) (See Fig. 1), between 2016 and 2020, can improve the effectiveness of models used to forecast weekly deaths by respiratory disease. A high-street retailer is a store located in a central hub of retail activity in a city, town or village.

## Use of novel approaches to investigate the added value of shopping behaviour data

It is important to note that establishing the capability of sales data to predict respiratory deaths is, unto itself, insufficient to prove their value to integrated disease models[35]. To warrant investment in the use of sales data in forecasting models, and the citizen privacy and commercial sensitivity issues which then must be surmounted, we must investigate the contribution that such data can offer uniquely, over and above other predictive variables[35]. To this end, in this study we employ an experimental approach specifically designed to identify the added value provided via non-prescription medication sales data in comparison with other covariates, leveraging recent advances in variable importance analysis[36,37]. In line with Hofman et al.'s recommendations on computational social science we construct baseline models prior to the addition of sales data, implementing a rigorous out-of-sample testing regime in order to ensure generalizability, with appropriate attention to both prediction and explanation tasks[38]. As detailed in the methods §4, and introduced in the results §2.2 and 2.5, two comparative models are created by attaching a range of independent variables (input features) for each of England's 314 LTLAs, derived from multiple datasets outside of sales data (the week of the year, weather, temperature, socioeconomic deprivation, age/population level, demographics, housing and land use). A further weekly dependent variable (output feature) is also attached to each LTLA, reflecting respiratory deaths in the authority for that week. We refer to these comparative models as PADRUS (Prediction of Amount of

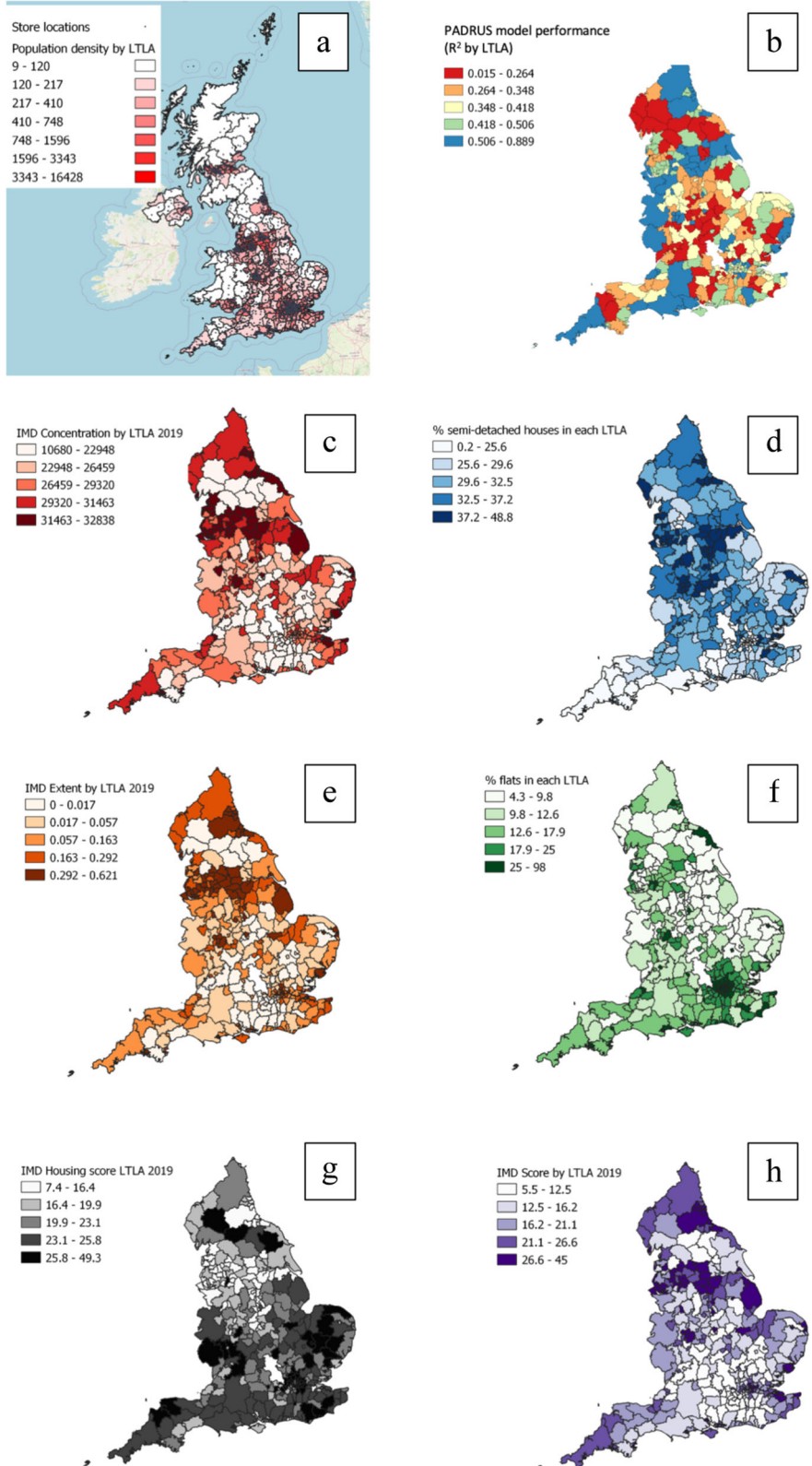

**Fig. 1 | Store, performance and traditional variable's importance maps.** Maps showing the United Kingdom and England divided by 314 Lower Tier Local Authorities (LTLAs) showing by LTLA **a** store locations and population density **b** the PADRUS model's performance by $R^2$, and the traditional static variables with the most impact on the model according to Model Class Reliance (MCR) **c** Index of Multiple Deprivation (IMD) concentration **d** Percentage of semi-detached houses **e** IMD extent **f** Percentage of flats **g** IMD housing score **h** IMD score.

Deaths by Respiratory disease Using Sales) and PADRUNOS (Prediction of Amount of Deaths by Respiratory disease Using No Sales). PADRUS has additional weekly behavioural input features created from shopping data attached to each local authority, encoding medication purchasing patterns across the LTLAs local population each week. These variables are derived from high-resolution sales data sourced from a national UK high-street retailer, including both relative and absolute weekly sales levels of a variety of of non-prescription medications bought for managing respiratory symptoms. Predictive models are then trained and assessed to identify an optimal forecasting mechanism based on this dataset. We then apply Model Class Reliance (MCR) analysis to ascertain variable importances; which variable inputs are the most important to the models' predictions.

MCR is a relatively new approach that allows researchers to examine the impact of variables each time a machine learning model is run, across all models that perform equally well, known as the 'Rashomon set'[39] (See Fig. 2).

No single execution of a predictive model can be guaranteed to provide a full explanation of the underlying phenomena it is modelling - in fact, far from it, with forecasting models that produce the same levels of accuracy often employing underlying variables in significantly different ways[36,40]. Such instability of explanation is due in part to the complex interactions that occur between variables and the mutual information that can be shared between them, which can serve to occlude confounding variables, and over-state the importance of correlated features[37]. Only through interrogation of a range of competing explanations (i.e. well performing models) can the utility and necessity of any individual input feature be rigorously assessed. With automatic enumeration of all competing, equally valid explanations intractable in most cases, MCR provides a mechanism to find the minimum impact a variable may have on a model (MCR−) and the maximum impact a variable may have (MCR+), by calculating the levels of usage of variables across competing explanations. MCR builds on permutation for a single model[41], computing the permutation feature importance bounds MCR- and MCR+. This can help identify a variable's absolute requirement (MCR−) and its maximal utility (MCR+) for forecasting efforts, rather than arbitrarily selecting a value between the two calculated from a single execution of a model, randomly selected from the Rashomon set. We also employ Group-MCR to test the importance of different variable categories as assessing which data inputs are most important can be difficult due to multicollinearity, extensive shared information and non-linear interactions occurring between variables[40].

By comparing the accuracy of PADRUS's weekly predictions of deaths by respiratory disease in each local authority against the baseline and comparative model (PADRUNOS), and through examination of variable permutation importance bounds, computed by MCR, we investigated the extent to which improvements are dependent on the integration of non-prescription medication sales data.

## Results
### National-level exploratory models using 5 million transactions logged by loyalty cards from November 2009 and April 2015
To determine forecast horizons, the length of time between when predictions are made and the actual date when the prediction is to occur, and to establish if linear relationships existed (see Supplementary Fig. 1), initial models were developed using sales and outcomes across the whole of England and Wales using 3-31 day intervals between the last day of the weekly sales aggregate (sales are collected Wednesday to Tuesday) and the day of reported deaths (deaths are registered on a Friday)(see Table 1, and Supplementary Fig. 2). Using over 5 million transactions from November 2009 to April 2015, weekly sales figures were totalled for England and Wales to make predictions between 7th December 2009 to 13th April 2015. Across this time period the maximum weekly deaths from provisionally registered

respiratory disease (as the underlying cause) in England and Wales was 3521 and the minimum 868, with 378,230 respiratory deaths recorded in total. This exploratory modelling used a separate data-set, covering a different time period, and available prior to acquiring the data used in the following local-level forecasting models. Input sales features were created from total weekly sales of cough, dry cough, mucus cough, decongestant and throat healthcare products, reflecting internal product categories used by the retailer and recorded through point-of-sale logging systems in stores. Regressors predicting weekly deaths from respiratory disease using sales data 17 days in advance achieved the strongest results, producing an out-of-sample $R^2$ of 0.81 on the held-out test data (30%). When predictions were made 24 days in advance, results were still strong (RMSE 224, $R^2$ 0.77), but performance was notably weaker when predicting within 10 or fewer days, or 31 or more days in advance. Number of days in forecast horizons are determined by weekly deaths being registered on Fridays, and the sales data week beginning on Wednesday and ending on Tuesday.

### Local-level forecasting models including those using over 2 billion in-store sales transactions with store location from March 2016 to March 2020
To examine the relationship between input variables and respiratory deaths at local authority levels (LTLAs) three models were created and are referred to as the baseline, PADRUS and PADRUNOS. All three models were non-linear, and utilised a random forest regressor (allowing for subsequent MCR analysis[39]), with parameters optimized via a time series cross-validation regime using a grid-search (see §4), and results given on held-out test data (30%). A random forest regressor is a non-linear machine learning algorithm which constructs a combination of decision trees in order to produce predictions[41,42]. Based on the results in Table 1 for national-level exploratory modelling, models used features constructed based on a 17 day forecast horizon for each of the 314 LTLAs in England between 18th March 2016 and 27th March 2020. In addition sales data lagged at 24 days were included as strong results were also seen for predictions with this time lag (See Table 1 and Supplementary Fig. 2). In the experimental design only England was included in the model due to data availability. Further explanation is given in the methods section.

The baseline model was constructed from two variable inputs consisting of week number, reflecting the seasonal nature of ILIs, and LTLA population over 65, reflecting the known vulnerability of this demographic to respiratory disease[43]. Models including only week number had an $R^2$ score of 0.007 on 30% test data and therefore this further variable of population over 65 was added to create a tougher-to-beat baseline. For the dependent variable (output feature) the maximum weekly registered deaths in any LTLA from respiratory disease in England was 219 and the minimum 0, with deaths 671,046 assigned in total. Note however, that these total figures are inflated due to weekly figures between 1 and 5 in an LTLA being suppressed, and assigned as 5 to ensure de-identification of any individual. It is noted that an anomaly in the data occurs on the 6th January 2017 where registered deaths are erroneously low. This has been left in the dataset to assist ease of future replicability including any generated statistics. As shown in Table 2 further described in section §2.5, despite its simplicity the baseline model showed reasonably high predictive power, forecasting weekly respiratory deaths in each LTLA with an $R^2$ of 0.71 (RMSE 4.00, MAE 2.78). Alongside, Root Mean Squared Error (RMSE), which takes into account the distribution of error magnitudes, Mean Absolute Error (MAE) was also calculated in order to provide a more interpretable measure of average error magnitude[44].

The PADRUS model was next constructed, utilising 56 features derived from sales, weather, demographic, environmental and socio-economic data (please see §4.2 for full details of independent variables). Using over 2 billion in-store sales transactions with store location from March 2016 to March 2020, sales data used was

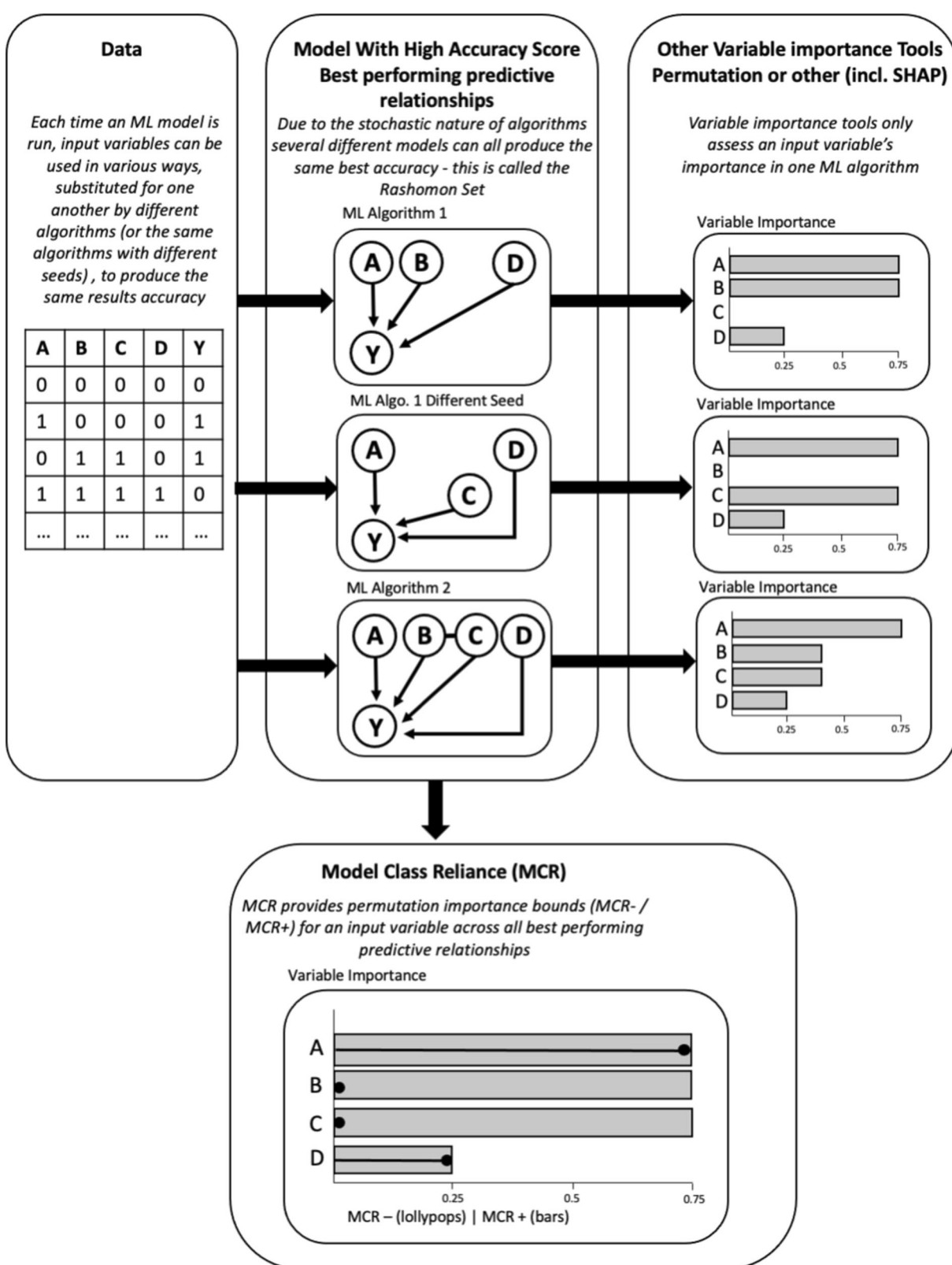

**Fig. 2 | Diagrammatic representation of the difference between other variable importance tools and MCR.** Figure abbreviations: MCR (Model Class Reliance), ML (Machine Learning), SHAP (SHapley Addictive exPlanations). NB, permutation importance relays the expected increase in error after a variables contents are randomly shuffled, and SHAP applies weighted linear regression to determine the importance of each variable based on Shapley values, as derived from game theory.

composed of units sold for a variety of self-medication products across 2354 retail stores in England during the time period. March 2020 was the last month used in the model as after this date the general sales at the retailers' stores were shut-down due to UK government lockdown coming legally into force on 26th march 2020, and the public told to stay at home on the 23rd of March 2020[45]. This meant the model could not generalise to the pandemic period, trained on data prior to the closing of shops, because of this huge economic and environmental

shift. Sales data variables were created from both the total sales from the week ending 24 days before the day of the prediction, and the total sales from the week ending 17 days before the prediction. These are two seven day non-overlapping periods (see Supplementary Fig. 2). We subsequently refer to these as variables with a 17 or 24 day lag. The forecast horizon for the PADRUS model was 17 days.

The PADRUS model significantly outperformed the baseline model achieving an $R^2$ of 0.78 (RMSE 3.42, MAE 2.39), producing significantly higher levels of predictive accuracy forecasting. In addition to the strong out-of-sample results, we note further the extremely high in-sample fit results, with PADRUS model producing an $R^2$ of 0.86 when modelling training data (again please refer to Table 2 for full results).

## Variable importance results

Permutation feature importance analysis was applied to PADRUS, relaying the expected increase in error after a features contents are randomly shuffled (see Supplementary Table 1 for full details). The size and age of an LTLA's population was identified as being the most important driver in producing model predictions, with mortality rates from respiratory disease being higher in older demographics. However, these factors were closely followed in importance by sales data features, and specifically the relative proportion of cough medication purchases. After these top ranked features, of next greatest importance to the model's predictions were IMD concentration (Index of Multiple Deprivation is a measure of relative socioeconomic deprivation across the country) and weather features, both having greater impact on forecasting than decongestant sales, and housing related features. IMD is the official measure of relative deprivation for areas in England, one of the statistics reporting levels of deprivation is IMD concentration which is the population weighted average of the ranks of a local authorities area's most deprived Lower Super Output Areas.

Permutation importance, while still being informative, is limited in its explanatory power, and feature importance analysis for PADRUS was hence also examined via SHAP (see Fig. 3). SHAP carries out a more complex calculation of feature importance, with the Kernel Explainer SHAP tool used in this analysis applying weighted linear regression to determine the importance of each feature based on Shapley values, as derived from game theory[46]. SHAP values were calculated on a random sample of 1000 output data points from an instance of the PADRUS model. SHAP values relay the extent to which the feature impacts on the model's output, and whether the impact increases or decreases the prediction. Results again reported that the proportion of the > 65 population as most important to the model's predictions, closely

**Table 1 | Results from exploratory linear regression models predicting weekly registered deaths from respiratory disease in England and Wales (min 828, max 3521), using over 5 million transactions from November 2009 to April 2015**

| Forecast Horizon | Results on National level data | | | |
|---|---|---|---|---|
| Days between sales & respiratory deaths | Results on training set 70% | | Results on test set 30% | |
| | Root Mean Square Error | $R^2$ | Root Mean Square Error | $R^2$ |
| 3 days | 137 | 0.86 | 286 | 0.62 |
| 10 days | 134 | 0.86 | 258 | 0.69 |
| 17 days | 146 | 0.84 | 203 | 0.81 |
| 24 days | 165 | 0.79 | 224 | 0.77 |
| 31 days | 191 | 0.72 | 270 | 0.66 |

**Table 2 | Results from experimental models predicting registered deaths from respiratory disease 17 days in advance using over 2 billion in-store sales transactions with store location from March 2016 to March 2020**

| Model | Months of the year included in results | Results on training set (approx. 70%) for 314 LTLAs | | | Results on test set (approx. 30%) for 314 LTLAs | | |
|---|---|---|---|---|---|---|---|
| **Experimental Design Predicting Deaths in English Local Authorities 18/03/2016 to 27/03/2020** | | RMSE | MAE | $R^2$ | RMSE | MAE | $R^2$ |
| Baseline Random Forest Regressor | All months | 3.41 | 2.4 | 0.78 | 4.00 | 2.78 | 0.71 |
| PADRUS Random Forest Regressor | All months | 2.74 | 2 | 0.86 | 3.42 | 2.39 | 0.78 |
| PADRUNOS Random Forest Regressor | All months | 2.87 | 2.08 | 0.85 | 3.61 | 2.49 | 0.76 |
| | | Results on training set for the 41 LTLAs with limited suppression | | | Results on test set for the 41 LTLAs with limited suppression | | |
| PADRUS Random Forest Regressor | All months | 4.27 | 3.32 | 0.85 | 5.87 | 4.37 | 0.70 |
| PADRUNOS Random Forest Regressor | All months | 4.52 | 3.53 | 0.83 | 6.5 | 4.8 | 0.63 |
| PADRUS Random Forest Regressor | Oct, Nov, Dec, Jan, Feb, Mar | 4.78 | 3.7 | 0.85 | 6.58 | 4.84 | 0.66 |
| PADRUNOS Random Forest Regressor | Oct, Nov, Dec, Jan, Feb, Mar | 5.11 | 3.94 | 0.83 | 7.43 | 5.47 | 0.57 |
| PADRUS Random Forest Regressor | Oct, Nov, Dec, Jan, Feb | 4.85 | 3.75 | 0.85 | 6.03 | 4.74 | 0.69 |
| PADRUNOS Random Forest Regressor | Oct, Nov, Dec, Jan, Feb | 5.24 | 4.04 | 0.82 | 6.86 | 5.31 | 0.60 |
| PADRUS Random Forest Regressor | Nov, Dec, Jan, Feb, Mar | 4.96 | 3.82 | 0.84 | 6.71 | 4.93 | 0.65 |
| PADRUNOS Random Forest Regressor | Nov, Dec, Jan, Feb, Mar | 5.33 | 4.11 | 0.82 | 7.64 | 5.64 | 0.55 |
| PADRUS Random Forest Regressor | Nov, Dec, Jan, Feb | 5.08 | 3.91 | 0.84 | 6.13 | 4.83 | 0.69 |
| PADRUNOS Random Forest Regressor | Nov, Dec, Jan, Feb | 5.54 | 4.28 | 0.82 | 7.07 | 5.52 | 0.58 |
| PADRUS Random Forest Regressor | Dec, Jan, Feb, Mar | 5.17 | 3.97 | 0.84 | 6.96 | 5.09 | 0.64 |
| PADRUNOS Random Forest Regressor | Dec, Jan, Feb, Mar | 5.57 | 4.3 | 0.82 | 7.92 | 5.85 | 0.54 |
| PADRUS Random Forest Regressor | Nov, Dec, Jan | 5.11 | 3.92 | 0.84 | 6.33 | 5 | 0.66 |
| PADRUNOS Random Forest Regressor | Nov, Dec, Jan | 5.7 | 4.39 | 0.8 | 6.94 | 5.38 | 0.59 |
| PADRUS Random Forest Regressor | Dec, Jan, Feb | 5.42 | 4.16 | 0.84 | 6.37 | 5.05 | 0.67 |
| PADRUNOS Random Forest Regressor | Dec, Jan, Feb | 5.95 | 4.61 | 0.81 | 7.36 | 5.79 | 0.56 |

*Listed in order of variable importance*

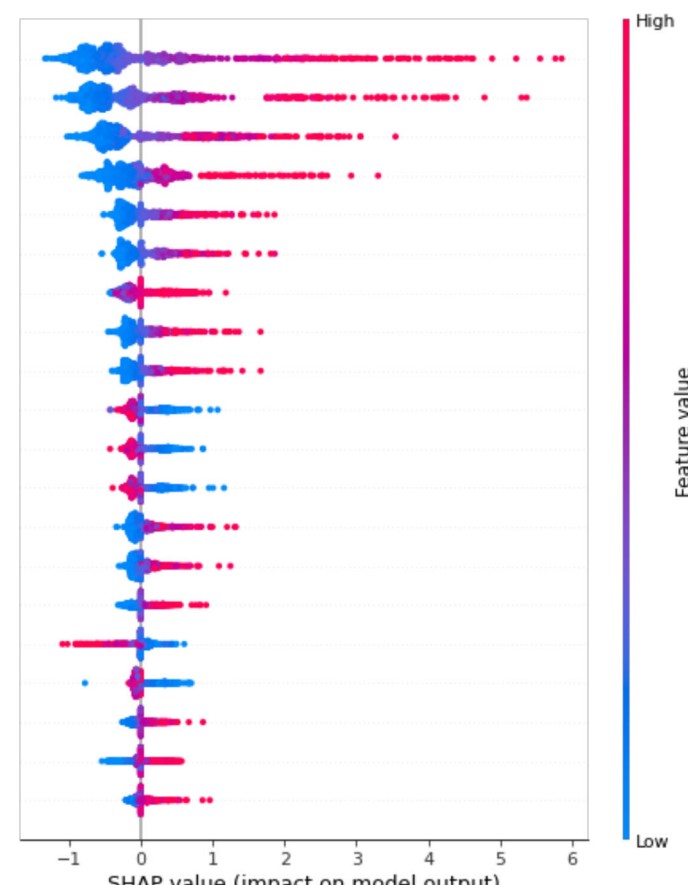

*N.B. Below 0 the variable negatively contributes to the value of prediction and above 0 the variable positively contributes to the value of the prediction. For example, higher population over 65 means higher respiratory deaths, whereas higher minimum temperature means lower respiratory deaths*

**Fig. 3 | SHAP analysis for feature importance assessment for the PADRUS model.** The Kernel Explainer SHAP tool was employed to calculate feature importance by applying weighted linear regression based on Shapley values derived from game theory. The tool was applied to the training data set on an arbitrary instance of the PADRUS model run on a random sample of 1000 data points.

followed by size of younger populations in an LTLA. However, this was again followed in importance by sales data, specifically all cough, and dry cough medicine sales figures with a 24 day lag. Next was IMD concentration, cough and dry cough medicine sales with a 17 day lag, and thereafter features created from weather data.

While potentially indicative of the important role that sales data might play in respiratory disease death predictions, standard permutation importance and SHAP analysis still only reflects the workings of a single arbitrary instance of the PADRUS model, drawn at random by the optimiser from the Rashomon Set of all equally well-performing models. As such, MCR was finally applied to understand the boundaries of this Rashomon set. MCR builds on permutation for a single model, computing the permutation feature importance bounds (MCR-, MCR+) for an input variable across all instances of PADRUS; calculating the minimum and maximum impact a variable could have on the predictions across all instances of the model (See Fig. 2). This initial MCR analysis evaluated each feature individually for its importance, for example the importance of 'minimum temperature'. The minimum impact a variable can have (MCR-) is represented on the MCR plots by the 'lollipops' and the maximum impact a variable can have is represented by the 'bars'.

The permutation importance bounds (MCR-, MCR+) returned by MCR (see Fig. 4, and Supplementary Table 1 for full details) indicate moderate differences in the way independent variables can be used in combination to forecast respiratory deaths. MCR indicated

unequivocally that the size of populations in the three age categories remain the most important factor, and that these were irreplaceable by any other variables in making optimal predictions (with an MCR-higher than any other variable sets MCR+). Population over age 65 was the most dependable feature for predictions with the highest MCR-.

While overall IMD (socioeconomic deprivation) scores for an LTLA had high importance in some models, this was not the case across the Rashomon set, and their relatively low MCR- scores indicated that they could be replaced by other variables (this excludes IMD concentration which consistently remains important). In contrast the features with highest MCR- scores, indicating that they were the most strongly required variable following population age and size, were features derived from cough medicine sales, with cough medicine sales with a 24 day lag outperforming a 17 day lag. Subsequent features with relatively high MCR- scores compared to other remaining variables, were IMD concentration, decongestant sales with a 17 day lag, minimum temperature, maximum temperature, average temperature, week of the year, and decongestant sales with a 24 day lag. Variables besides the aforementioned showed consistently low MCR- scores, yet several retained stronger MCR+ scores which indicates that these input features were likely broadly interchangeable with each other.

When a large set of independent variables are used as inputs to a predictor, assessing the importance of each feature to achieving a model's optimal accuracy can be challenging, and results are difficult to interpret in unison. To address this we applied Group-MCR, which

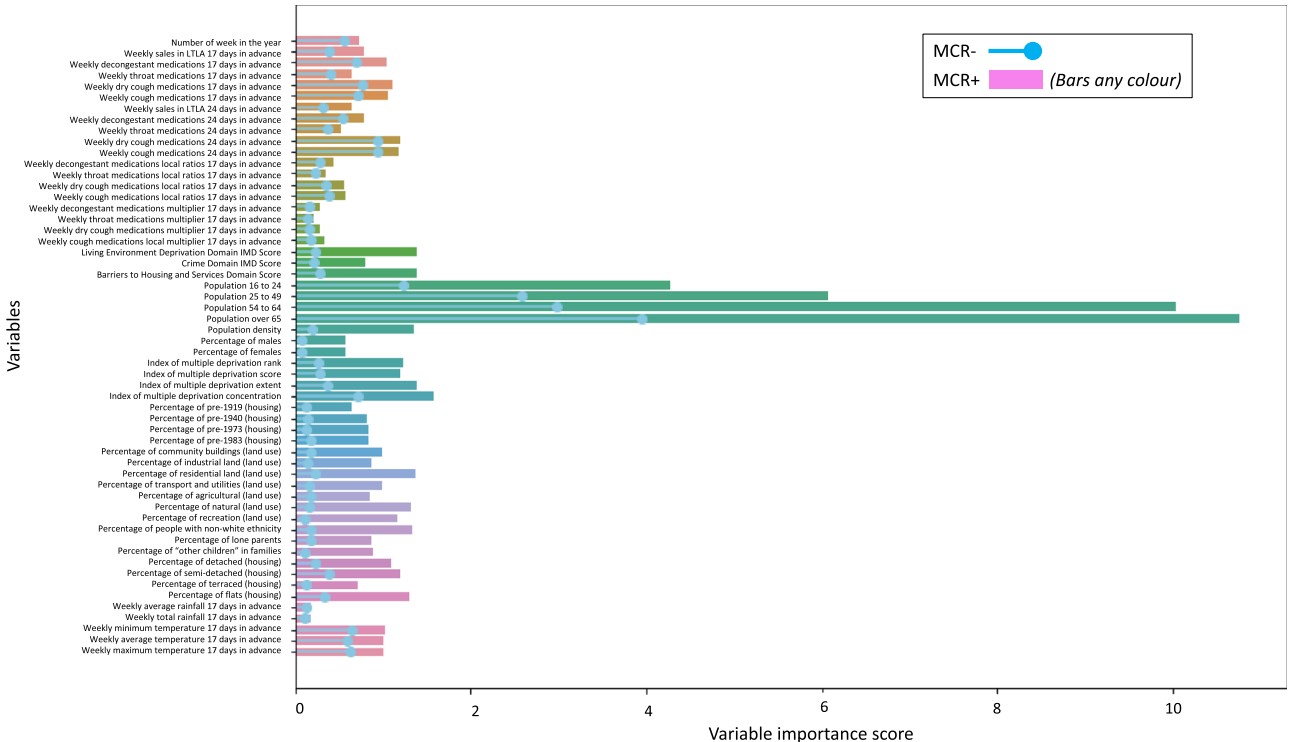

**Fig. 4 | Model Class Reliance (MCR) results for PADRUS.** MCR computes the permutation feature importance bounds (MCR−, MCR+) for an input variable across all instances of PADRUS; calculating the minimum and maximum impact a variable could have on the predictions across all instances of the model. MCR was applied to the PADRUS model trained on 45,844 data points.

computes the importance of a category of features in concert, calculating the minimum and maximum impact variable groups could have on the predictions across all instances of the model[40]. Individual variables and their groups are in Supplementary Table 2. Results for Group-MCR are reported in Fig. 5. Features related to population age are substantially the most important group of features (MCR− 22.84, MCR+ 26.05). When grouped, sales features, however, are second in importance to achieving optimal predictions (MCR− 11.99, MCR+ 12.87). Following local population age and size, non-prescription medication sales are most central to models producing optimal 17-day forecasts of respiratory deaths in local authority regions.

Sales features provided substantially higher permutation importance bounds than the third in importance, IMD (MCR- 3.68, MCR+ 5.63). In fourth and fifth position, variable groups housing (MCR- 2.31, MCR+ 4.05) and land use (MCR- 1.95, MCR+ 3.87) had very similar MCR scores. For weather (MCR− 2.45, MCR+ 2.54) in sixth position of importance and week number (MCR− 0.56, MCR+ 0.73) the least important in eighth position, there was only a small difference between MCR- and MCR+ scores. This implies weather and week number consistently have the same permutation importance across all model instances. Demographics was in seventh position (MCR− 0.68, MCR+ 2.14). The high MCR−/+ for grouped sales features means that there are no models within the Rashomon set that can achieve their predictive scores ($R^2$ 0.78) without this set of variables (Fig. 5).

To compare the assessment of variable importance using the method of removing each set of grouped variables and creating new models with the usage of Group-MCR, please refer to the results from additional models presented in Supplementary Table 3.

### Comparison to models without sales data
MCR analysis indicates that the prediction accuracy produced by PADRUS could not be achieved without the inclusion of sales data variables. However, MCR in its current form, cannot tell us the exact

accuracy increases gained through their inclusion - only that they are essential to those gains made. To compute the difference in accuracy, a further model PADRUNOS was used, constructed from the same variable set as PADRUS but omitting features derived from sales data. To derive the PADRUNOS model, a random forest regressor was again optimised using a time series cross-validation grid-search to predict weekly deaths from respiratory disease 17 days in advance for the 314 LTLAs.

Results were notable in that the PADRUNOS model achieved a strong $R^2$ of 0.76 (RMSE 3.61, MAE 2.49) on the 30% held-out test dataset (20,410 data points), despite the absence of non-prescription medication sales (see Table 2 for full details). While the improvements made by PADRUS over PADRUNOS detailed in Table 1 were significant (p-value < 0.001, using Bonferroni corrected Wilcoxon signed-rank tests), accuracy increases were relatively small (0.02 $R^2$). Wilcoxon test results comparing the models results on held-out 30% test set were p-value < 0.001 across all result scoring metrics, 95% confidence intervals $R^2$ (−0.023, −0.01), MAE (0.033, 0.11), RMSE (0.077, 0.189). Group-MCR was again applied to investigate how PADRUNOS was compensating for the loss of sales data (see Fig. 5 for a side-by-side comparison with PADRUS). Results show that the PADRUNOS model became even more reliant on the grouped variables of age (MCR− 37.91, MCR+ 41.21). The dynamic variable, weather (MCR− 7.55, MCR+ 7.54) increased in its relevance, becoming the second most important feature used for forecasting. The related variable, week number (MCR− 2.04, MCR+ 2.15) which encodes seasonality also increased its importance. All other grouped variables increased their minimum and maximum permutation bound scores, apart from land use (MCR− 1.15, MCR+ 3.85) where the MCR- decreased.

### Assessing critical prediction periods and suppression
We observed that optimal forecasting models are dependent on sales data; yet broader analysis appeared to indicate that the accuracy gains

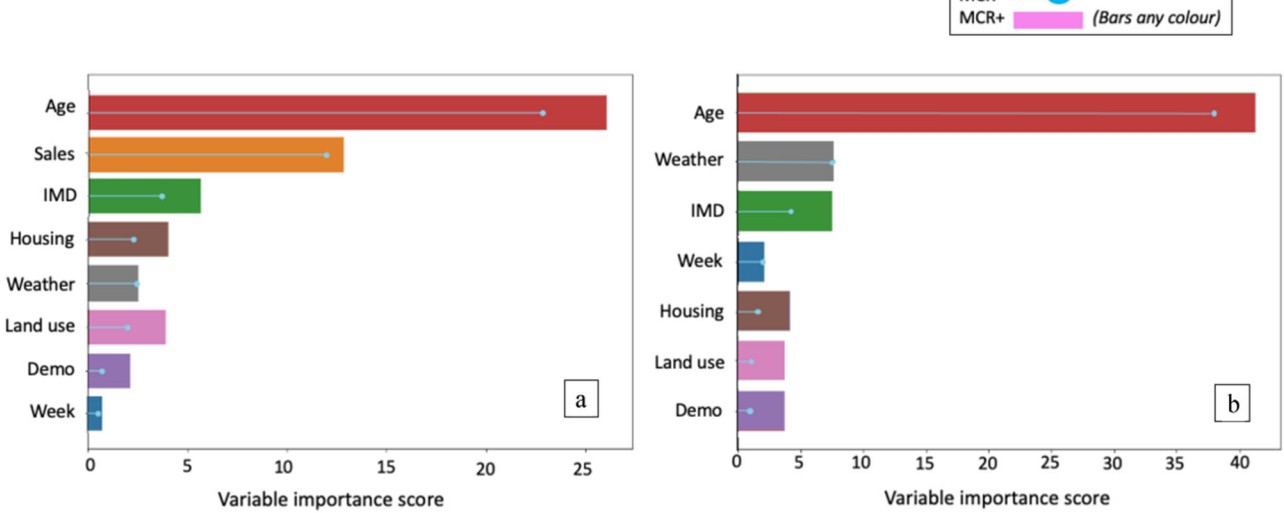

MCR- (lollipops showing minimal impact variable has on predictions) | MCR+ (bars showing maximum impact variable has on predictions))

**Fig. 5 | Group-MCR results for PADRUS and PADRUNOS. a** PADRUS and **b** PADRUNOS. Group-MCR assesses variable group effects on random forest classifier predictions, computing importance bounds (MCR−, MCR+) for input variable groups across all instances of the model. Group-MCR was applied to the PADRUS model trained on 45,844 data points. 'Demo' stands for demographics. NB difference in scale of graph.

arriving through their inclusion are relatively small. To understand this apparent contradiction, and explain how sales data can simultaneously appear to be both critical to predictions while only providing limited improvements in accuracy, we examined weekly time-series forecasting results across LTLAs for both models (plotted in Fig. 6). Forecasting graphs illustrate that although both PADRUS and PADRUNOS follow similar predictive patterns between 2016 and 2020, PADRUS is better at capturing key spikes in respiratory death rates that are underestimated by PADRUNOS. It is here that inclusion of sales data obtains accuracy gains, over and above reliance on seasonal indicators and weather-related variables. Comparing PADRUS and PADRUNOS prediction scores ($R^2$, MAE and RMSE) by week (Fig. 7, Supplementary Table 4) further highlights how the inclusion of sales data increases the PADRUS model's predictive accuracy especially in winter months. Accuracy gains are hence limited over the whole year, but occur in periods most important to early warning systems.

A second key observation is obtained through examination of how results vary across individual LTLA areas (see Fig. 1 and Supplementary Table 5 for more details). These indicate that prediction accuracies are being impacted by inclusion of suppressed data. Death counts below 5 within an LTLA were suppressed and reported as 5 prior to receipt of the data. When predictions were limited to the 41 LTLA areas with data that contained no suppressed weeks, it became clear that the PADRUS model was not only performing significantly better than PADRUNOS, but with clearer effect sizes. For unsuppressed LTLA data, PADRUS achieved an $R^2$ of 0.70 (RMSE 5.87, MAE 4.37) compared to PADRUNOS' $R^2$ of 0.63 (RMSE 6.50, MAE 4.80). As shown in Table 2 and Supplementary Fig. 3, this reflects an improvement in $R^2$ of 0.07, indicating that suppression was simplifying the true prediction task, to the advantage of models without behavioural sales data.

Further, when analysis of predictions in these LTLAs is constrained to seasonal periods of higher respiratory disease (e.g. November - February) the improvement in performance of the PADRUS model ($R^2$ 0.69, RMSE 6.13, MAE 4.83) further increases in comparison to models without sales data ($R^2$ 0.58, RMSE 7.07, MAE 5.52). Winter periods, when vulnerability to respiratory disease is most prevalent, reflect weeks when suppression is less likely but also when regressors have the most scope for error due to higher potential incidence. Depending on the exact start and end month chosen to

capture these critical periods for forecasting, inclusion of sales data increases $R^2$ between 0.09 to 0.11 (again full results are detailed in Table 2 and in Supplementary Fig. 4 a point plot figure illustrates these results).

Additional models were generated to compare the inclusion and utilisation of LTLA identifiers for calculating historical death averages by area. The corresponding results can be found in Supplementary Table 3. The model relying solely on week numbers and LTLA identifier inputs exhibited considerably lower accuracy ($R^2$ 0.67, RMSE 4.21, MAE 2.93). Furthermore, the addition of the LTLA identifier variable alongside the 56 variables used by PADRUS had minimal to negligible impact ($R^2$ 0.78, RMSE 3.43, MAE 2.39).

**Assessing economic regional factors on prediction accuracy**

Variation in the accuracy of the models' predictions on the weekly deaths from respiratory disease may not only be influenced by suppression, other independent variables of an LTLA area could impact the results (See Fig. 1). In consideration of this, additional stratification of the results by the Index of Multiple Deprivation (IMD) was conducted to assess the impact of the economic circumstances of the population. Using IMD concentration, as it was shown to be the most important IMD measure to the model in the variable importance analysis, results on the test set (approximately 30%) for the 314 LTLAs were stratified by IMD concentration interquartile ranges. This analysis revealed both models PADRUS and PADRUNOS produced predictions with higher accuracy in areas with higher concentration levels of deprivation, with PADRUS out-performing PADRUNOS across all interquartile ranges and this out-performance in R2, MAE and RMSE increasing as the deprivation increased (Table 3).

## Discussion

Results of this study evidence that the inclusion of non-prescription medication sales data commonly bought for managing respiratory symptoms, alongside traditional variables, can contribute to increases in forecasting accuracy for respiratory deaths. Models corroborate prior work[22–25], but also provide potential insights into why contradictory results exist within the literature[19,32,34]. While initial results in this study achieved only relatively small effect sizes in across-the-year accuracy gains (with an increase of 0.02 $R^2$ over models without sales

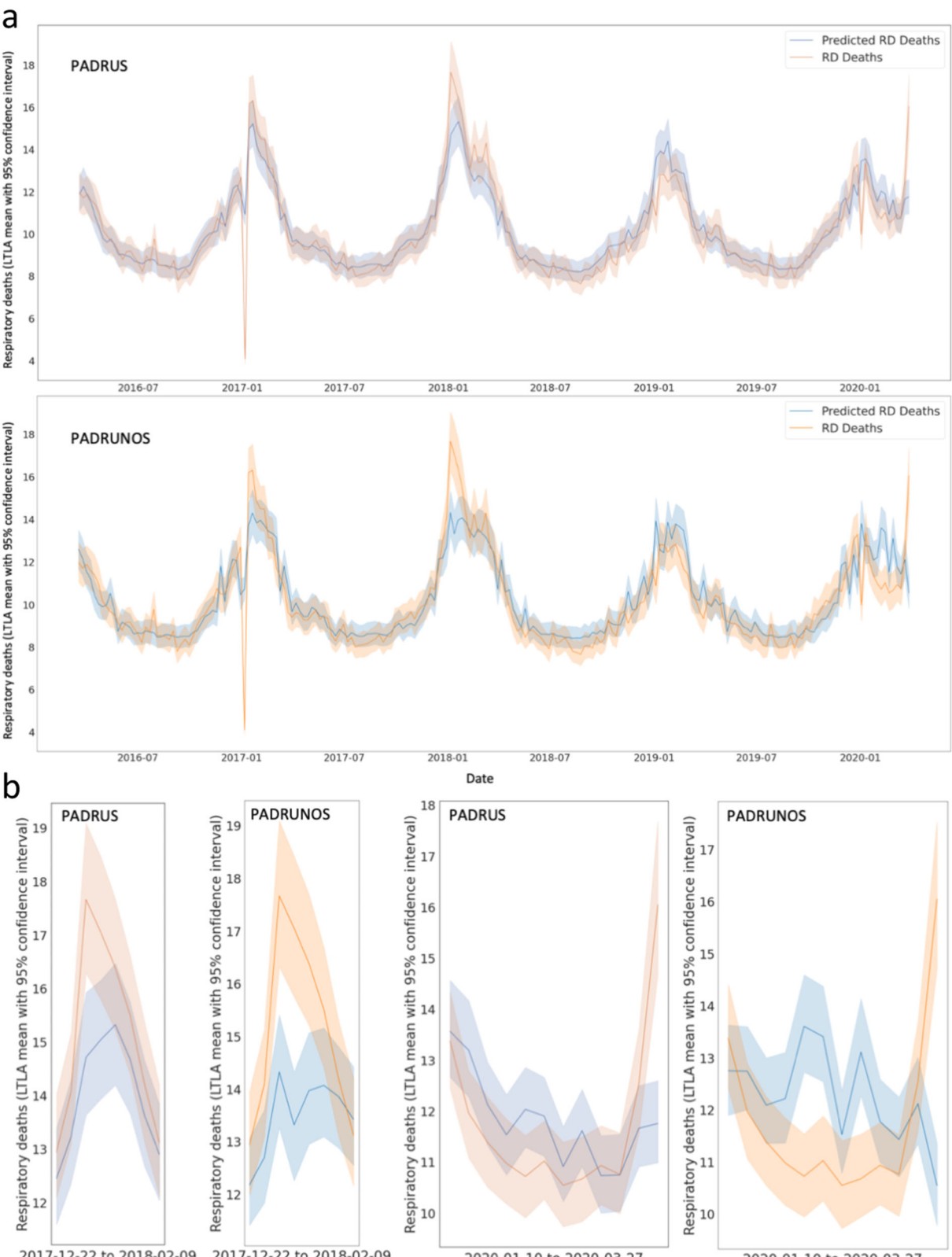

**Fig. 6 | Line Plot of PADRUS and PADRUNOS predictions and actual number of respiratory deaths. a** Panels show results are from 66,254 datapoints (45,844 training datapoints, 20,410 testing datapoints) of 211 weekly deaths from respiratory disease for 314 English Local Authorities from 18th March 2016 to 27th March 2020. **b** panels provide magnified visualisation of the proximity of the blue area outlines (predictions) to orange areas (targets) in PADRUS and PADRUNOS at specific peak and trough periods.

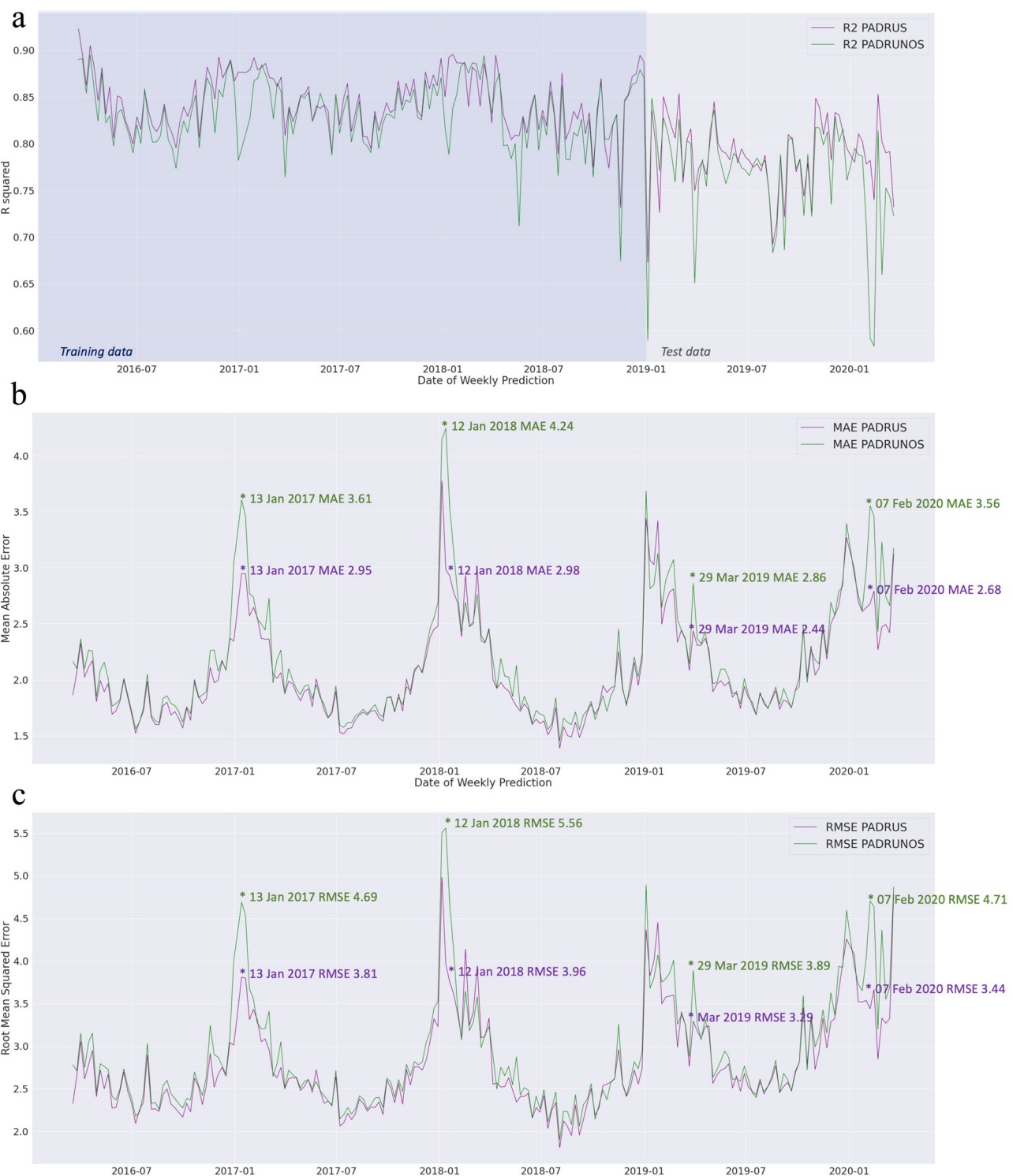

**Fig. 7 | Line Plots showing how the inclusion of sales in the PADRUS model changes. a** R² (coefficient of determination), **b** MAE (mean absolute error) and **c** RMSE (root mean squared error) predictive scores especially in the winter months. (Three data points have been left out where error scores were higher to create a high granular view to compare models. These data points were the anomaly 2017 week 1, 2020 week 1, and 27th March 2020 the first week of the UK lockdown). Results are from 66,254 datapoints (45,844 training datapoints, 20,410 testing datapoints) of 211 weekly deaths from respiratory disease for 314 English Local Authorities from 18th March 2016 to 27th March 2020.

data), further analysis revealed that improvements were being constrained by over-simplification of the modelling task due to suppressed data, and a lack of focus on periods when incidence of respiratory disease is the most volatile. The models ability to accurately capture small death counts was limited by the suppression process whereby all weekly death counts below 5 were reported as 5. When these issues were attended to, the gains made through inclusion of sales increased markedly, with out-of-sample forecasting increasing in predictive performance by 0.11 ($R^2$) when models included behavioural sales data.

**Table 3 | Test-set (approx. 30%) results from PADRUS and PADRUNOS Random Forest Regressor models predicting weekly registered deaths from respiratory disease in England for 314 LTLAs stratified by Index of Multiple Deprivation Concentration (IMD) Interquartile Ranges (IQR) in LTLAs**

| LTLA IMD IQR Range (IMD concentration) | PADRUS | | | PADRUNOS | | |
|---|---|---|---|---|---|---|
| | R2 | MAE | RMSE | R2 | MAE | RMSE |
| 0–25 (10,680 – 24,164) | 0.61 | 2.35 | 1.72 | 0.59 | 2.4 | 1.73 |
| 25–50 (24,164–28, 327.76) | 0.68 | 3.06 | 2.19 | 0.64 | 3.22 | 2.27 |
| 50–75 (28,327.76 – 30,999.650) | 0.71 | 3.56 | 2.53 | 0.67 | 3.76 | 2.65 |
| 75–100 (30,999.650–32,838.16) | 0.81 | 4.37 | 3.11 | 0.77 | 4.72 | 3.32 |

In addition to confirming the influence of age and population size, variable importance analyses (in the form of MCR/Group-MCR) showed that optimal predictions were achieved using medication sales data, notably the cough sales medication feature with a 24 day lag (with IMD concentration and temperature also providing utility). The importance of cough sales recorded 24 days in advance shows the potential for predictions from sales data to have forecast horizons long enough in length to inform the forward planning needed for allocating health resources. Why sales features with 24 day lag appear to have a higher variable importance to the PADRUS model than those with a 17 day lag needs further investigation, however the length of this duration indicates that lagged medication sales may not be predicting hospitalisation in the people that purchase them - but of a growing risk within a community. This risk could be an increase in infectious diseases affecting the respiratory system, or another environmental change which is exacerbating and/or increasing chronic lung conditions. Confirmation of this hypothesis, however, would require accompanying individual level studies.

MCR could also enable decision making in choosing alternative groupings of the more traditional variables included in the model. By informing us which of those variables were most consistently critical to the models for producing high accuracy rates, and which variables might be interchangeable, our MCR analysis of PADRUS could help to select variables that are easier or more affordable to access, are most up-to-date, and would be most accepted by the public to use in surveillance systems for respiratory disease. In PADRUS a number of variables are gathered from the Office for National Statistics (ONS) census 2011 (Supplementary Table 2), the census takes place every 10 years collecting data on all households in England and Wales, and MCR reveals these are possibly conveying similar information to the model as variables from the government statistics on land use from 2018, and housing age data from the Valuation Office Agency 2020 (Supplementary Table 2). This is particularly important as model inputs related to housing made up 3 out of the top 6 most important static traditional variables excluding population age (See Fig. 1 and Supplementary Table 1).

MCR analysis suggests that the gains occurring due to inclusion of sales data are due to weather/ temperature variables being unable to sufficiently compensate for when disease incidence veers away from seasonal norms. A relationship between temperature and ILIs is well noted in the literature, but there is no current consensus on how colder weather might cause the seasonal increase in respiratory disease[47]. Moreover, when disease incidence diverges from historical patterns due to feature drift, in this use-case a rapid change in the relationship between seasonable variables and deaths, predictive models are unable to respond quickly enough. An example is the 2017/18 influenza season in the UK, which was unexpectedly characterised by co-circulation of both influenza A (H3N2) and influenza B. Vaccination regimes were unable to respond accordingly to this situation, leading to a significantly large number of respiratory outbreaks and

increased hospital admissions[48]. In PADRUNOS, seasonal/temperature variables were unable to adapt to this change in causal relationship (or 'feature drift'), whereas in PADRUS the relationship between non-prescription medication sales and deaths remained far more stable. For disease surveillance under adaptation, variables that respond to disease prevalence, and obtained from direct empirical observation of human behaviour, may provide a more stable and reactive forecasting mechanism.

Socioeconomic deprivation in an area measured through IMD was found to be an important variable group to both PADRUS and PADRUNOS model, with regional differences in levels of IMD concentration influencing the models' accuracy scores. Both models performed better where deprivation was higher, but this increase in accuracy was higher for PADRUS which also consistently achieved superior prediction scores (see §2.6, Table 3, Fig. 1). Why inclusion of sales would increase accuracy gains in areas with higher deprivation needs further investigation. A possible hypothesis is populations in more deprived areas are more likely to have long term lung diseases such as asthma and chronic obstructive pulmonary disease (COPD)[49–52], and therefore if the sales of cough medicines indicate there has been an environmental change which can exacerbate chronic lung conditions, increases in deaths will be higher in these at risk communities. This theory highlights how sales data could potentially help in an early warning system, with at risk patients sent alerts when predictions from models including sales data show a predicted increase in deaths, especially outside the seasonable norms. Those at risk patients could be sent text, push notifications or email messages to advise them to adhere to medications. The latter are already digital technologies being trialled using smart inhalers and smartphones[53,54]. General practitioners and staff in primary care healthcare settings could also be informed to ensure the care teams of patients, and where appropriate families, with conditions such as COPD have extra help adhering to their medications where there may be barriers using digital aids, or accessing prescription medications[55]. Hopefully these preventative measures could help reduce predicted deaths, yet the early warning system could also help in patient flow modelling, predicting future patient arrivals for healthcare facilities, and hospital planning, particularly for winter periods where high patient admissions are experienced[56–58].

This study considered four continual years of weekly retail sales data in 314 LTLAs across England, providing finer resolution evidence for previously hypothesised relationships between 1. sales of cough and decongestant medications and rates of ILI[30] and 2. sales of cough medicines and peaks in hospital admissions for respiratory illness two weeks later[33]. No significant evidence for the relevance of other non-prescription medications, such as throat remedies, was found. Due to the temporal granularity of the dataset made available to this work, it was also possible to examine a range of the models results constrained to particular periods across the year (see Table 2 for a full list of these). PADRUS provided significant improvements over PADRUNOS (and accounting for Bonferroni corrections) but sales data proved most valuable to the periods covering October to February. This is likely due to both the increased magnitude of deaths over this period, and hence the higher potential volatility of mean squared error that can occur - periods that forecasting models are of most utility to decision makers.

Predictive mechanisms able to adapt to changing circumstances are, of course, also highly relevant to the surveillance of pandemics such as COVID-19, where new unpredictable diseases occur due to such factors as antigenic shifts. While analysis of this period was outside the scope of the data available for this work, a small crossover period does occur (up to March 27th). At this point the PADRUS model predicts that deaths from respiratory disease were set to increase, a response to increased non-prescription medication sales it observes, whereas PADRUNOS model, relying on seasonal variables, predicts a fall. There is insufficient data covering this period, however, to test

whether this forecasting advantage would continue - with reported panic buying of medical supplies and a shift into online shopping, feature drift may have weakened the relationship between sales and hospitalisations. However with COVID-19 still prevalent worldwide and continuing to regularly defy the normal seasonal trends of ILIs, further research is required to investigate if the potential holds to underpin early warning mechanisms. Extended research could assess whether new data available during the pandemic such as COVID-19 test, case and mortality data, and mobility data[59], could create accurate models utilising sales data. Noted, the feature drift of reduced sales post and between varying lockdowns would need to be addressed. Research into using sales data to predict deaths in COVID-19 could help to confirm this study's findings, offering many more peaks and valleys in disease prevalence to assess a model's ability to generalise to new data.

This study is limited by its use of in-store purchases from one UK high street retailer despite its wide coverage with thousands of stores across England, and the vast majority of the population being within <10 min from a store. Sales data may represent a biased sample of the population due to the demographic characteristics and socio-economic status of the type of consumers who can access, afford and choose this retailer. While the use of a single lens into medication sales does not impact the validity of forecasting results, and indeed likely means that reported forecasting accuracies represent a lower bound of potential forecasting improvements, representativeness of results are difficult to examine. Any real-world forecasting model based upon a single source of data is necessarily biased towards the demographic it featured most heavily. Furthermore, areas of the country where a retailer has greatest presence (which in this case tend to be urban, rather than rural locales) will feature most predominantly within a regression, and consequently receive higher probabilities of accurate predictions. This may be attended to by normalisation/weighting towards representative areas/customer types, but further investigation is required.

We also note that an anomaly in the data occurred on the 6th January 2017 where registered deaths were very low for the time period. Public holidays affect the process of death registrations[60], and the anomaly may have occurred due to Christmas Day in 2016 occurring over a weekend. A final limitation, and a potentially fruitful route for further research, lies in the need to undertake data linkage between individual customer data and clinical outcomes if we are to unpack drivers of model accuracy. This would require a participatory cohort study of shoppers willing to undertake data donation, and careful sampling across demographics, geospatial regions and of those who have and have not experienced respiratory disease.

This study provides evidence of the potential value in integrating national longitudinal shopping data into respiratory disease forecasting models acting at local levels. Analysis indicates value in including sales data in predictive models in order to identify irregularities outside seasonal norms, and in periods where deaths from respiratory disease were most prevalent and varied. With COVID-19 now endemic in the population the potential of shopping data to improve forecasting is even more valuable for aiding healthcare decision-making. The prospect of increasing the capability of sales data to improve forecasting models for respiratory disease is promising; the PADRUS model is at the lower bounds of possible predictive accuracy with an array of limitations and improvements that future research could address.

## Impact Statement
The results of this study show the potential of sales data to aid timely forecasts in disease surveillance which could be of use to healthcare decision making, however, this potential would be subject to public health authorities being able to access commercial sales data in real-time. Careful negotiations with large retailers would need to be conducted, with due consideration to commercial sensitivity, and how the

costs for the data analytics, systems infrastructure, and employee time and skills would be accounted for. The ability to achieve the latter would vary across different countries and geographic regions dependent on data quality both in health and retail, technology infrastructure and the population skill-sets. Despite the variation across different environments, there will be both opportunity and a financial cost for incorporating sales data into disease surveillance systems that must be balanced. Therefore, cautious evaluation is essential to ensure that models can adapt to both environmental and consumer changes, ensuring the longevity of systems reliant upon them.

It must be noted that events can occur that generate sharp changes in human behaviour, directly impacting the effectiveness of sales-based models. For instance, the accuracy of the PADRUS model would be highly affected by circumstances such as government lockdowns preventing in-store sales. Additionally, current models may not account for item shortages resulting from panic buying. Further research is necessary to develop information systems capable of monitoring and providing warnings of rapid feature/distributional shifts of this nature. Simultaneously, care must be taken to integrate relevant moderating variables (e.g., introducing a variable encoding item availability) in response to such changes. These shifts, whether major or gradual, can be caused by various factors such as evolving economic conditions, product range change, shop closures, or shifts in disease symptoms due to different virus variants altering the medications bought in response. The development of AI (Artificial Intelligence) explainability tools that track changes in variable importance across time series data, and can identify feature drift, could be crucial in maintaining model accuracy and informing policymakers about new disease symptoms and possible implications for the model's applicability.

## Methods
### Open-source software used
We used the following open-source software in the data collection and analysis:

1. Open source PostgreSQL 13: https://www.postgresql.org
2. Matplotlib: https://matplotlib.org/
3. MCRForest: https://github.com/gavin-s-smith/mcrforest
4. Numpy: https://numpy.org/
5. Pandas: https://pandas.pydata.org/
6. Seaborn: https://seaborn.pydata.org/
7. Scikit-learn: https://scikit-learn.org/
8. SHAP: https://shap-lrjball.readthedocs.io/en/latest/index.html

### Shopping, mortality and Socio-demographic data
Data was collected for this study through working partnerships with: the UK National Health Service (NHS); a UK commercial high street retailer; and via open source data providers. Data sources and descriptions used in this study can be seen in Supplementary Table 2, which lists the data used for both dependent and independent variables. In Supplementary Table 2, the table column 'Research relating variable to respiratory disease' cites the journal papers providing scientific evidence why each variable type was included.

Data underpinning medication sales variables originates from two data-sets made available to the project by a national UK high street retailer, incorporating: 1. a 20% sample of timestamped commercial sales transactions (including over 5 million transactions of non-prescription medications for cough, decongestant and throat) from England and Wales recorded through 2,702,449 loyalty cards from November 2009 to April 2015; and 2. a data-set of all in-store weekly commercial sales transactions in England (over 2 billion) labelled by store location, covering the period between March 2016 to March 2020. Commercial sales data was sales units of all in-store transactions with store location only; no sales transactions were linked to individual customers, and no personal data was used in this study. Other

independent variables used in the models are derived from: English indices of deprivation 2019[61]; Nomis official census and labour market statistics[62]; Housing age data from the Valuation Office Agency 2020[63]; ONS Census 2011[64]; Land use in England from 2018 live tables from the Department for Levelling Up, Housing and Communities and Ministry of Communities and Local Government[65]; and Weather ERA5 data from the European Centre for Medium-Range Weather Forecasts[66].

Data was collected or aggregated to the geographic regions of LTLAs (Local Tier Local Authorities). The LTLAs boundaries used in this study are the 314 local governmental areas across all of England as structured in 2019. LTLAs were used because, as geographic areas used by government data, they would enable direct inclusion of a wide range of demographic, environmental and socioeconomic data previously found to be significant to the risk of death from respiratory disease (Supplementary Table 2). LTLAs were chosen over other spatial divisions, including census areas MSOAs (Middle Layer Super Output Areas) and LSOAs (Lower Layer Super Output Areas), to find a balance between limiting data suppression of deaths done to maintain data de-identification, and maintaining a high enough level of spatial granularity for data to have a significant impact on predictions.

The output variable for the models, respiratory deaths in each of the England's 314 Local Authorities (LTLAs), originates from two datasets supplied by the UK Office of National Statistics (ONS) and the National Commissioning Data Repository (NDCR) containing: 1. details of total weekly registered deaths from respiratory disease (as the underlying cause)(ICD 10 coding: J00 - J99) in England and Wales from 7th December 2009 to 13th April 2015[67]; and 2. ONS data on all weekly registered deaths from respiratory disease (ICD 10 coding: J00 - J99) by 314 LTLAs in England from 18th March 2016 to 27th March 2020. The health data in this study was used under the terms of usual practice for research as defined in the UK Policy Framework for Health and Social Care Research, with the study designed to investigate the health issues in a population to improve population health[68]. To keep health data de-identified and extractable from the NDCR, death counts below 5 within an LTLA were suppressed and reported as 5 prior to receipt. Suppression was between 20% and 25%.

Open source data availability for UK LTLAs outside of England was limited, and consequently analysis was for local-level forecasting was restricted to 314 LTLAs in England alone. Even with this restriction other variables under consideration for inclusion such as mobility data, search engine trends, temporal pollen, traffic, and air pollution counts, could not be included as there was no access to these datasets at this level of geospatial granularity or for the 4 year period within the time limits of the project. Data matching, the time series of available target predictions and available dynamic datasets of sales and weather, gave the dataset its time-frame of the 18th March 2016 to 27th March 2020 for weekly deaths by respiratory disease.

## Exploratory modelling / National-level Analysis

Preliminary exploration of the national UK relationship between sales data and respiratory deaths was undertaken before modelling at local levels to better understand potential forecast horizons. For this analysis a distinct preceding time period was considered, using ONS data on weekly registered deaths from respiratory disease (as the underlying cause) in England and Wales from 7th December 2009 to 13th April 2015[67] and medication sales data drawn with forecast horizons of 3, 10, 17, 24, 31 days (forecast horizon refers to the time lapse between the date of registered deaths being predicted and the date of reported sales). Independent variables consisted of 'week number' (1 to 52) and features created from commercial sales data, including absolute weekly sales units for: cough medication, dry cough, mucus cough, decongestant and throat healthcare products, reflecting the data partner's internal product categories. Sales features were aggregated by week from over 5 million over-the-counter transactions recorded by 2,702,449 loyalty cards. Data was temporally stratified into a training

(70%) and test set (30%), and out-of-sample performance examined against the held-out test data via RMSE (root mean squared error) and $R^2$ (coefficient of determination). Several linear regression models were trained (one for each forecast horizon at which sales data was engineered) and evaluated to test for the optimal forecast horizon in days between sales and registered deaths (Table 1). Results then guided subsequent feature engineering stages of fully cross-validated local-level modelling experiments, as described in the following section.

## Local-level respiratory-death forecasting

The central experiment of this study, with forecasting at local-level, was structured into two phases: modelling and variable importance analysis. Phase 1 investigated a range of random forest regressors to identify an optimized reference model, PADRUS, for forecasting respiratory deaths 17 days in advance (as indicated by exploratory national-level modelling). All available data used in this modelling covered the period between 18th March 2016 and 27th March 2020, with variables reflecting hypothesised associations with deaths from respiratory disease (see Supplementary Table 2 for a full feature list). In order to provide comparison two other models were generated - baseline, using just week number and LTLA population over 65, and - PADRUNOS, which used all variables apart from those derived from sales data. Phase 2 of the analysis sought to explain the impact of the inclusion and importance of different variables on the models' predictions, including commercial sales data. Phase 2 applied the novel variable importance tool MCR for random forest regressor[37,40].

**Feature engineering.** All raw data was aggregated to LTLA spatial resolutions across England, with 314 LTLAs being assigned weekly values for the dependent variable (target feature) and each of the 56 independent variables (input features). The data used for the dependent variable was all registered deaths from respiratory disease (ICD 10 coding: J00 - J99) by LTLA on a weekly time basis. Independent features used by models can be categorised into dynamic or static variables. Dynamic variables had a forecast horizon of 17 days. Static features, which were assumed to be constant over the period of analysis, were themselves grouped into variables sets including: demographics, indices of multiple deprivation, age of population, housing and land use. Dynamic (temporal) features were grouped into: week number, commercial sales, weather-related. These groupings were used for the Group-MCR (Model Class Reliance), and further details of 56 features were created can be seen in Supplementary Table 2 which lists any data manipulation from the original source (e.g. use of averages/percentages/aggregation approaches).

**Feature engineering for sales data.** Sales data variables were created from both the total sales from the week ending 24 days before the day of the prediction, and the total sales from the week ending 17 days before the prediction. These are two seven day non-overlapping periods. We refer to these as variables with a 17 or 24 day lag. Online sales were not included, and all data was pre-aggregated to store level. Based on this pre-aggregation, no individual customer information was known, nor could be reconstructed. Features derived from sales data were included as both absolute and relative values for each medication category, with relative values further represented as both a proportion of that weeks store sales, as well as in a form normalised by overall national sales, to account for changing footfall.

Complex data manipulation was needed for feature creation from commercial sales. Sales from the 2354 stores, distributed as they are across England, had to be assigned to each of the 314 English LTLAs. To help achieve this we first introduce some notation. The total number of unit sold across the whole UK in a particular week for product category, $c$ is defined as: $sales_{UK,c}$.

While total sales across the country can simply be counted, the number of units bought by customers from any particular LTLA, $sales_{L,c}$, cannot, given that people will travel across LTLA boundaries to purchase medications. To estimate the proportion of sales that arise from customers living in a particular local authority a store catchment model must be be established. We implement a simple Gaussian kernel-based approach that allows us to infer the 'influence' each store has over an LTLA, and then use that to attribute sales to LTLAs proportionately.

We establish a two-dimensional Gaussian with variance, $\sigma_s^2$, over each store, $s$, with a mean, $\mu_s$, centred at the store's longitude and latitude. $\sigma_s^2$ was varied depending on a theorised catchment radius (metres) corresponding to a store's type (Regional Mall 15,000 m, Hospital / Transport Hub 10,000 m, Retail Park / Shopping Centre 8000 m, High Street / Supermarket 5,000 m, Health Centre/Community 3000 m). These catchment ranges reflect $2\sigma$ (95% of the kernel) and were devised in collaboration with the data provider. The influence a store, $s$, had over any point, $x$, can therefore be assigned as:

$$influence_s(x) = \frac{1}{\sigma_s \sqrt{2\pi}} e^{-(x-\mu_s)^2 / 2\sigma_s^2} \qquad (1)$$

The 'influence' the store had over an LTLA is then determined by examining the value of the store's probability density function at that LTLA's population weighted centroid. Due to the fact that Gaussian functions have an infinite range, a store will be assigned a non-zero influence for every LTLA across the UK but, in order to simplify analysis, the influence assigned to a point by each store for a given LTLA was zeroed if it fell below a threshold of 3.449937e-318. The 'total influence' of each store can then be calculated by summing the influences for the population weighted centroid of every LTLA, $L$, from the 314 available, $\mathcal{L}$:

$$total\ influence_s = \sum_{L \in \mathcal{L}} influence_s(L) \qquad (2)$$

The amount of sales that each store attributes to a given LTLA, $L$, can then simply be assigned as a proportion of the influence the store has over that LTLA compared to its total influence overall. If sales are then aggregated for each store then we can finally produce an estimate of category sales in that LTLA overall. Formally for any sales category, $c$, the absolute sales assigned to an LTLA, $L$, in particularly week, after aggregating all contributions of every store, $s$, is:

$$total\ sales_{L,c} = \left( \sum_{s \in S} \frac{influence_s(L)}{total\ influence_s} \times sales_{s,c} \right) \qquad (3)$$

Total sales in any given week, however, does not always represent an informative variable. This is due to the uneven geographical distribution of the retailer's stores across the country, and the different number of customer bases they serve. A more informative variable is the percentage of sales that a category, $c$, is contributing that week - in comparison to sales across all product categories, $\mathcal{C}$. To this end, we derive a local sales ratio variable for an LTLA, $L$, as follows:

$$sales\ ratio_{L,c} = \sum_{\substack{c' \in \mathcal{C} \\ c' \neq c}} \frac{total\ sales_{L,c}}{total\ sales_{L,c'}} \qquad (4)$$

Here $\mathcal{C}$ represents all product categories sold by the retailer (including but not restricted to non-prescription medication categories) and we are therefore normalising by overall sales in that LTLA. This prevents modelling bias towards regions where there is a higher population or number of retailer stores.

It is useful to compare the sales ratio occurring in an LTLA with the national average, as this allows to defend against both feature drift and seasonal fluctuations. To achieve this we define a local sales multiplier

variable, which allows us to assess whether non-prescription medication sales are proportionally more dominant in one region compared with another (making this feature more invariant over time):

$$local\ sales\ multiplier_{L,c} = \frac{sales\ ratio_{L,c}}{sales\ ratio_{UK,c}} \qquad (5)$$

These three features total sales, sales ratio and local sales multiplier make up the central sales variables within our models (with versions for cough medicines, dry cough medicines, decongestant medicines and throat medicines - products selected due to reports in prior literature associating a rise in their sales with an increase in respiratory illnesses[30,33]).

**Modelling procedure.** The analysis of registered deaths from respiratory disease was framed as a regression task with the amount of deaths predicted (the target) a continuous output integer variable, $y$. The modelling task was prediction of weekly respiratory deaths 17 days in advance for each of the 314 LTLA areas in England from 18th March 2016 to 27th March 2020. For all models (PADRUS, PADRUNOS, baseline) data was temporally stratified into the same training set (45,844 data points from 18th March 2016 to 28th December 2018), approx. 70%) and test set (20,410 data points from 4th January 2019 to 27th March 2020, approx. 30%)

Random forest regressors were trained and evaluated, with meta-parameters for the model optimised using a time series cross-validation (TSCV, 4 splits) and grid search to prevent over-fitting. TSCV is an alternative to k-fold cross-validation, which is applied to prevent temporal data leakage and the risk of over-reported accuracy results. TSCV iteratively applies a 'walk-forward' testing process, in order to robust evaluate a models' ability to generalise. Once optimal meta-parameters had been identified, the model was re-fit to the training dataset, and evaluated against the held-out test set. Three metrics were used to score the regression task: RMSE (root mean squared error), MAE (mean absolute error) and $R^2$ (coefficient of determination).

To assess whether the PADRUS model created was valuable, two baseline models were created. The first, baseline, was created using the same methodology; a random forest regressor optimized using a time series cross-validation grid-search on training data. The baseline performed the same task of predicting weekly respiratory deaths 17 days in advance for each of the 314 LTLA areas in England from 18th March 2016 to 27th March 2020. Data was stratified using the same splits on the training and test data (45,844 training data points, 20,410 test data points), and model predictions on the held out test data were scored using RMSE, MAE and $R^2$. The only difference between PADRUS and the baseline model is its feature inputs, with the baseline inputs consisted of only: the week number (1 to 52) for a dynamic input, and secondly the static input of the population of over 65s in each LTLA. The first feature was chosen because of the known seasonality of deaths by respiratory disease[47], yet used alone the prediction accuracy levels were very low. Therefore the second feature was chosen due to the assumption that the size and age of population would greatly influence the number of deaths, with 90% of deaths from respiratory disease in Europe occurring in those aged 65 and over[43]. The second comparative model, PADRUNOS, was formed in exactly the same way, and used the exact same features as PADRUS except any that related to sales data (see §4.4.5)

**Variable importance analyses.** In order to understand the impact of the different features driving predictions in the PADRUS model, we conducted a range variable importance analyses. First standard permutation importance was used to score of each feature in producing the model's predictions, relaying the expected increase in error after the a features contents are randomly shuffled. Second, a SHAP

(SHapley Additive exPlanations) Analysis[46] was conducted. SHAP carries out a more complex calculation of feature importance, with the Kernel Explainer SHAP tool used in this analysis applying weighted linear regression to determine the importance of each feature based on Shapley values, as derived from game theory[46]. Both tools were applied to the training data set on an arbitrary instance of the PADRUS model. The SHAP analysis is computationally expensive approach and therefore was run on two random samples of data points of 100 and 1000. This enabled a comparison of results between sample sizes to ensure a large enough sample had been evaluated.

By running these variable importance tools on one arbitrary model, misleading results can be given due to the stochastic nature of the models machine learning algorithms. This is a problem because different instances of an optimised model class can use different variables (features) and in different ways to achieve the same model predictive performance. To address the problem of standard variable importance tools only evaluating one instance of the PADRUS model, MCR (Model Class Reliance) was applied. MCR was developed by Fisher et al. to compute the feature importance bounds across all optimal models called the Rashomon set for Kernel (SVM) Regression (polynomial run-time)[36]. Smith, Mansilla and Goulding introduced a new technique that extends the computation of MCR to Random Forest classifiers and regressors[37]. MCR builds on permutation for a single model, computing the permutation feature importance bounds (MCR-, MCR+) for an input variable across all instances of PADRUS; calculating the minimum and maximum impact a variable could have on the predictions across all instances of the model (See Fig. 2). This initial MCR analysis evaluated each feature individually for its importance, for example the importance of 'minimum temperature'. It did not assess how important a dataset type used to create a number of the models' features was to predictions as a whole, for example 'weather'. Ljevar et al. introduced a Grouped Feature approach to MCR for random forest[40]. Group-MCR was created in order to calculate the effects of variable groups, measuring the importance of a collection of features together on the predictions of random forest classifiers[40]. In order to evaluate the importance of groups of variables, and their impact in concert on the PADRUS model, we apply for the first time Group-MCR to a random forest regressor. Group-MCR is achieved through a modification of the random forest MCR algorithm, which reconsiders the definition of Model Reliance to be with respect to a group of variables rather than a single variable[40].

**Modelling without sales data.** Although MCR can explain which variables need to be included in a model to achieve the maximum accuracy rates for predictions, it can not compute the difference in accuracy (the loss) if those variables were to be excluded from the model. In order to determine the loss in accuracy if sales data were to be left out of the model, the model PADRUNOS (Predicting the amount of deaths from respiratory disease using no sales) was created as a comparison to PADRUS. PADRUNOS was created inline with the baseline and PADRUS model methodology, as a random forest regressor optimized using a time series cross-validation grid-search on training data. MCR and Group-MCR was applied to PADRUNOS to calculate how the variables were used differently. The three metrics used to score the regression task were compared for both models PADRUS and PADRUNOS. These scores were examined overall, by week, by LTLA and by winter months. Wilcoxon signed-rank tests (two-sided) were used to compare the significance of model's predictive accuracy scores.

**Models for further understanding and reference.** In order to determine the loss in accuracy if each set of grouped variables were to be left out of the model, and to enhance understanding of how the MCR variable importance tool functions by comparison, separate random forest regressor models were created each omitting a different variable group (See Supplementary Table 3). A further baseline random forest regressor model was also created using the variables week number and LTLA identifier only, to compare the accuracy of predicting on yearly historical averages of deaths only. A final comparative model was then created by incorporating the LTLA identifier variable into a random forest regressor alongside the 56 variables used in PADRUS (See Supplementary Table 3). The latter aimed to assess whether the LTLA identifier itself provided any additional predictive information beyond the 56 separate features that offered subject-specific data about each LTLA.

### Ethics and data management

This work was carried out within an NHSX PhD Internship: Placement Programme[69] and continued post-placement partnership. After reviewing the project proposal, NHS England (formerly NHSX) determined that the NHS Research Ethics Committee was not required based on criteria[68,70,71]. Health data in this study was used under the terms of usual practice for research as defined in the UK Policy Framework for Health and Social Care Research[68], with the study designed to investigate the health issues in a population in order to improve population health. To keep health data de-identified and extractable from the National Commissioning Data Repository (NDCR), death counts below 5 within an LTLA were suppressed and reported as 5 prior to receipt. Health data was deleted following the study's conclusion. Ethics approval for the study was additionally obtained from the University of Nottingham ethics panel (Faculty of Social Sciences NUBS ethics review reference: 201829095).

Both sales and health data were aggregated weekly to Local Authority areas for an ecological analysis, and abided by the retailers privacy policy for data use, and the lawful GDPR basis for processing consumption data was encompassed by: 'the performance of a task in the public interest' (GDPR 6(1)(e)) and 'processing necessary for scientific research purposes' (GDPR 9(2)(j)). Transactions logged via loyalty cards were geographically aggregated to a national level (England and Wales) and aggregated temporally to a weekly sales total. Sales units of in-store transactions were tagged with store location only; no sales transactions were linked to individual customers, and no personal data was used in this study. Strict data management protocols were abided to at all times, as per the N/Lab data management plan[72] and the University of Nottingham's ISO/IEC 27001 certification.

### Reporting summary

Further information on research design is available in the Nature Portfolio Reporting Summary linked to this article.

## Data availability

The health data used in this study is not publicly available but can be requested via NHS England and Improvement NCDR[73]. The sales data used in this study is commercially sensitive and subject to strict access controls. To request route to access, please contact the Neo-demographics Laboratory (N/Lab) Director, James Goulding (james.goulding@nottingham.ac.uk). We will promptly provide guidance on engaging with the commercial retailer and endeavour to respond within one month. All other datasets are open source and can be accessed via the website links given in Supplementary Table 2.

## Code availability

The analysis code developed for this paper can be found online at https://github.com/nhsx/commercial-data-healthcare-predictions. This can also be accessed via https://github.com/LizzieHD/sales-predictions-respiratory/releases/tag/v.final[74].

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

## Acknowledgements

This work has been supported by the grant, The CIVIC Project: A Sustainable Platform for COVID-19 syndromic-surveillance via Health, Deprivation and Mass Loyalty-Card Datasets (EPSRC grant, EP/V053922/1)(E.D., J.G., G.S., G.L., H.M., and L.T.). Elizabeth Dolan is supported by the Horizon Centre for Doctoral Training at the University of Nottingham UKRI grant EP/S023305/1. We thank NHSX in supporting Elizabeth Dolan with an NHSX PhD Internship: Placement Programme which enabled this work.

## Author contributions

All authors contributed significantly to this work: E.D., J.G., G.S., and L.T. jointly conceptualised the research, and E.D., J.G., G.S., G.L., H.M., and L.T. developed the methodology. E.D., J.G., G.S., G.L., and H.M. were responsible for data collection. E.D., J.G., G.S., G.L., H.M., and L.T. collaboratively drafted the original manuscript. E.D., J.G., G.S., and G.L. created data visualisations. All authors edited and approved the final manuscript.

## Competing interests
The authors declare no competing interests.
