## [Peer Review File · Nature Communications]

Assessing the value of integrating national longitudinal shopping data into respiratory disease forecasting modelsEditorial Note: This manuscript has been previously reviewed at another journal that is not operating a transparent peer review scheme. This document only contains reviewer comments and rebuttal letters for versions considered at *Nature Communications*.

REVIEWER COMMENTS

Reviewer #2 (Remarks to the Author):

Thanks to the authors for their thorough and detailed response to comments from the previous review of the manuscript.

With regard to comparing sales data to mobility data, it is unfortunate that mobility data are no longer available via SafeGraph or other channels, but the authors' good faith effort to obtain mobility is noted and appreciated.

Clarification about why the sales data used in the paper are cutoff during the start of the COVID-19 pandemic is helpful. That said, this still leaves an important open question: is one actually better off incorporating sales data into models like this under the types of large distributional shifts (e.g., pandemics, recessions, etc.) that occur in the world?

One way to think about this is to compare it to Google Flu Trends. When Flu Trends was initially created, the novelty of using search to "predict the present" was celebrated, but over time distributional shifts led Flu Trends predictions to deviate substantially from actual influenza caseloads, to the point that the model was decommissioned. It seems likely that the same could end up happening here, so the key question is how one would know if/when to use predictions from a model that incorporates sales data over a model that omits it.

The paper does a lot to highlight cases in which sales data can be helpful during "critical prediction periods", but plays down periods where such a model could be harmful. A more balanced perspective could help policy and decision makers better weight the pros and cons of investing in and relying on such models.

In terms of methodology, there is some conditioning on the dependent variable in removing areas with low death counts throughout the timeframe of the dataset. What about instead incorporating a fixed effect by area via LTLA identifiers and/or auto-regressive death counts for earlier time periods by LTLA? This way (only) the training data would be used to avoid overfitting, and the model would essentially incorporate trends in historical averages when making forecasts. This itself would be an interesting and practical baseline to compare to, and would avoid any selection on the dependent variable. (See also Goel et al, 2010, which shows that such models can be surprisingly competitive: an auto-regressive model on influenza cases with up-to-date data was shown to perform nearly identically to a model based on search activity, at least during the time the study was conducted.)

Another methodological point that was raised in the previous reviews was the use of SHAP and MCR for variable importance, and the idea of complementing this with a simpler ablation study. The authors have done a good job in clarifying details of these sophisticated algorithms, but it would be nice to add an ablation study for three reasons. First, simply removing features and re-training is easy for a broad audience to understand. Second, it answers the very practical question: is it worth a modeler's time to track down and incorporate a certain feature or set of features in terms of predictive performance? Variable importance measures give a proxy for this, but it's an indirect one, and incapable of answering what the return on investment (in terms of R^2 , RMSE, or MAE) is to adding features to a model. Third, by presenting an ablation study first and comparing that with variable importance approaches, the authors would be able to highlight the virtues of these methods to detail what they capture that ablation studies don't, as pointed out in the response to previous reviews.

A smaller point is that Table 2 would be much easier to read as a figure.

Reviewer #3 (Remarks to the Author):

I think that the authors have done a very good and careful job in addressing my concerns expressed in my review report. I feel that the paper is novel, more readable and accessible in its current form and provides a strong showcase of the value of predictive modelling by ML and dynamically accumulating data. I am happy to support the acceptance of the paper.

Response to editors and reviewers document

Manuscript: "Assessing the value of integrating national longitudinal shopping data into respiratory disease forecasting models"

Please see PDF manuscript for highlighted changes. Highlights are numbered and the numbers referenced in our authors responses. Highlight is abbreviated to H, e.g. see H1.

Reviewer #2

Remarks to the Author:

1) Thanks to the authors for their thorough and detailed response to comments from the previous review of the manuscript. With regard to comparing sales data to mobility data, it is unfortunate that mobility data are no longer available via SafeGraph or other channels, but the authors' good faith effort to obtain mobility is noted and appreciated.

1) Author response: *We appreciate these comments.*

2) Clarification about why the sales data used in the paper are cut-off during the start of the COVID-19 pandemic is helpful. That said, this still leaves an important open question: is one actually better off incorporating sales data into models like this under the types of large distributional shifts (e.g., pandemics, recessions, etc.) that occur in the world?

2) Author response: This is an extremely good observation, and one that speaks to much of the continuing research we are currently engaged on with the NHS. There was obviously a distinct change in the expression of respiratory disease at the start of the COVID-19 pandemic, with incidence moving rapidly and acutely away from "business as usual". However, it is exactly in such circumstances (i.e. within virus diversification/hocks) that we have since found that incorporation of sales data seems to improve predictive performance the most - likely due to this data channels sensitivity to disease incidence (being a responsive variable), compared to a sharp drop in the predictive power of traditional seasonal modelling produced as outbreaks generate seasonally-atypical infection patterns.

However, we felt that the ability of medication data to respond to such rapid "feature drift", and the predictive advantage that might ensue during outbreaks, was too bold a claim to be made without further detailed study at this time. In particular we felt such claims would require more involved comparisons to be made with statistical outbreak models (e.g. MERMAID), highly relevant to such periods but beyond the scope of the initial work. As such a decision was made to concentrate first on the potential fine-tuning value of medication datasets in periods when seasonal models were still functionally relevant.

We also did not want to over-emphasize the value of medication data in such periods without further attention to other variables affected by outbreaks. For example in lockdowns there can occur a sharp reduction in store opening/availability - and a corresponding additional dependent variable is required to accommodate for this. This remains an important point, and one that we agree with the reviewer was underemphasized in the paper. We thank them for their comments here - and to highlight that new data streams are not unto themselves panaceas, given the risk of feature/distributional shifts, we have added this additional text to the discussion:

(H5) *"It must be noted that events can occur that generate sharp changes in human behaviour, directly impacting the effectiveness of sales-based models. For instance, the accuracy of the PADRUS model would be highly affected by circumstances such as government lockdowns preventing in-store sales. Additionally, current models may not account for item shortages resulting from panic buying. Further research is necessary to develop information systems capable of monitoring and providing warnings of rapid feature/distributional shifts of this nature. Simultaneously, care must be taken to integrate relevant moderating variables (e.g., introducing a variable encoding item availability) in response to such changes."*

3) *One way to think about this is to compare it to Google Flu Trends. When Flu Trends was initially created, the novelty of using search to "predict the present" was celebrated, but over time distributional shifts led Flu Trends predictions to deviate substantially from actual influenza caseloads, to the point that the model was decommissioned. It seems likely that the same could end up happening here, so the key question is how one would know if/when to use predictions from a model that incorporates sales data over a model that omits it.*

3) Author response: The reviewer raises a really interesting issue of a change in the relationship between independent and output variables (again this typifies feature drift), and how models must be reactive to these changes if they are to be deployable in the real world (although we believe this applies to all models rather than just sales-driven ones). In the case of Google Flu trends this feature drift, and ultimate model failure, was partly due to the fame of the model itself, and rapidly changing search behaviours within the population. While not immune to feature drift PADRUS is likely to be more resilient here, as it is self-medication sales are purely driven by individuals attending to illness.

However, the failure of Google Flu Trends was also due to model overfitting - for example search terms such as "High School Basketball" have been highlighted as being used within it's predictions, despite having no relevance at all to the phenomena being modelled. Within PADRUS such overfitting is highly defended against, both with rigorous walk-forward cross validation - but also because the model uses *theoretically grounded input variables*, limited to medications directly related to symptoms (and not, as would be analogous to the overfitting of Google Flu trends, sales of items such as "sun cream," despite such items selling highly in periods with less disease incidence). This theoretical grounding is again good practice in real-world machine learning.

Nonetheless, the reviewer's attention to feature drift remains a fantastic observation. Even though PADRUS is likely more resilient due to our attention to overfitting risk, and with symptom expression constant across most respiratory diseases, such drift is possible - indeed it has been observed in COVID-19 where virus diversification/evolution gradually shifted the relationship between incidence and self-medication purchase choices (from, for example, cough mixture earlier in the pandemic to decongestants in later phases). As such we have added the text below reiterating these issues.

(H5) *"These shifts, whether major or gradual, can be caused by various factors such as evolving economic conditions, product range change, shop closures, or shifts in disease symptoms due to different virus variants altering the medications bought in response. The development of AI explainability tools that track changes in variable importance across time series data, and can identify feature drift, could be crucial in maintaining model accuracy and informing policymakers about new disease symptoms and possible implications for the model's applicability."*

4) *The paper does a lot to highlight cases in which sales data can be helpful during "critical prediction periods", but plays down periods where such a model could be harmful. A more balanced perspective could help policy and decision makers better weight the pros and cons of investing in and relying on such models.*

4) Author response: We agree a balanced perspective is key, and certainly do not wish to claim self-medication data is any panacea to the challenges of modelling disease incidence, but rather that it is a useful potential new tool in disease surveillance. Having said that, we did not find periods where PADRUS's outputs could be considered harmful - and we made pains to identify periods where PADRUS offered improvements and when it did not to actually try and provide balance (given that the addition of extra data streams to any early warning system comes at a logistical cost, so attention must be given to where it is actually of value). Nonetheless, we very much take the reviewers point of clarifying this, and as such have added the following sentences to the discussion:

(H6) *"Despite the variation across different environments, there will be both opportunity and a financial cost for incorporating sales data into disease surveillance systems that must be balanced. Therefore, cautious evaluation is essential to ensure that models can adapt to both environmental and consumer changes, ensuring the longevity of systems reliant upon them."*

5) *In terms of methodology, there is some conditioning on the dependent variable in removing areas with low death counts throughout the timeframe of the dataset. What about instead incorporating a fixed effect by area via LTLA identifiers and/or auto-regressive death counts for earlier time periods by LTLA? This way (only) the training data would be used to avoid overfitting, and the model would essentially incorporate trends in historical averages when making forecasts. This itself would be an interesting and practical baseline to compare to, and would avoid any selection on the dependent variable. (See also Goel et al, 2010, which shows*

that such models can be surprisingly competitive: an auto-regressive model on influenza cases with up-to-date data was shown to perform nearly identically to a model based on search activity, at least during the time the study was conducted.)

5) Author response: To attend to this, we have created two further models (both for reference and improved understanding of potential fixed effects), adding results to table 2 (**H1**). The first is an additional baseline model including only the variables of week number and area by LTLA identifiers (as recommended by the reviewer). The second model adds encoded LTLA identifiers to the complete set of 56 variables (see additional text in methods **H7**). The inclusion of a fixed indicator such as an LTLA identifier can of course encode a wide range of variables within it, and these were originally omitted due to the fact that assessing variable importances, a key goal of the study, is highly challenging in mixed-effect models (inclusion of an LTLA identifier risks masking the predictive strength of other key variables that one wants to investigate, such as deprivation levels, housing types, density, etc.). However, we recognize that their inclusion, and understanding their impact on prediction accuracies, remains an interesting addition.

Slightly surprisingly, inclusion of LTLA indicators into models did not significantly improve model accuracies (in either our baseline or full models). In fact our baseline model actually fell in accuracy with their addition, and on reflection this may have been for two key reasons: 1. with a wide set of 56 variables already, there was likely much shared information between these and LTLA identifiers (i.e. high collinearity rendering improvements negligible) 2. With a large number of LTLAs, introducing greater complexity through fixed effects, meant a corresponding loss of degrees of freedom when estimating other parameters, especially given limited data in each LTLA (curse of dimensionality). Nonetheless, please see the inclusion of these results in table 2 (**H1**). Additional text on this has been added to the results (**H4**).

Finally, and in reference to inclusion of auto-regressed death counts, we were asked by our data partners at the NHS not to include ongoing updates on registered deaths as an input feature due to the fact that in deployment situations such figures could not be reliably used. We were told that this is due to the fact that such figures can often be very inaccurate, with lengthy delays in registration of deaths being commonplace due to logistical/process challenges (with the data used as ground-truth in our models having been updated months after their labelled dates, in order to ensure they held accurate figures). We note that during the pandemic, temporary systems were put in place to record deaths due to COVID-19 both faster and with more accuracy, and we mention this case in the discussion on page 15.

6) Another methodological point that was raised in the previous reviews was the use of SHAP and MCR for variable importance, and the idea of complementing this with a simpler ablation study. The authors have done a good job in clarifying details of these sophisticated algorithms, but it would be nice to add an ablation study for three reasons. First, simply removing features and re-training is easy for a broad audience to understand. Second, it answers the very practical question: is it worth a modeler's time to track down and incorporate a certain feature or set of features in terms of predictive performance? Variable importance measures give a proxy for this, but it's an indirect one, and incapable of answering what the return on investment (in terms of

R², RMSE, or MAE) is to adding features to a model. Third, by presenting an ablation study first and comparing that with variable importance approaches, the authors would be able to highlight the virtues of these methods to detail what they capture that ablation studies don't, as pointed out in the response to previous reviews.

6) Author response: We agree with the reviewers recommendation, and have now added an ablation study where each set of variable types (grouped variables based on the data source) is removed and a new model created/tested for reference. We have added this addition in the methods **(H7)** and have incorporated the results into table 2 **(H1)**. Happily, the results echo those within our previous MCR analysis, with sales data showing continued utility (see **H1** table 2). Additional text has been added to the results to allow the reader to make this comparison between variable removal techniques and MCR for variable importance analysis **(H2)**.

7) A smaller point is that Table 2 would be much easier to read as a figure.

Author response: Taking into consideration this feedback, we have created a figure representing the majority of the results shown in table 2 and which relays the differences in results between months for those LTLAs with limited suppression. This figure should help to digest the results of table 2 more easily. The figure also illustrates the changes in R², MAE and RMSE across different periods between the models PADRUS and PADRUNOS, and is included in **supplement number 7** (referenced in the paper at **H3**).

Reviewer #3

Remarks to the Author:

I think that the authors have done a very good and careful job in addressing my concerns expressed in my review report. I feel that the paper is novel, more readable and accessible in its current form and provides a strong showcase of the value of predictive modelling by ML and dynamically accumulating data. I am happy to support the acceptance of the paper.

1) Author response: We thank Reviewer 3 for their comments, time and effort.

REVIEWERS' COMMENTS

Reviewer #2 (Remarks to the Author):

Thanks to the authors for yet another round of thorough and detailed responses to previous comments. I appreciate the time and effort that went in to addressing these concerns and am now in support of the paper for acceptance.

The additional text that has been added to qualify claims and state limitations or areas of possible concern is helpful.

With regard to the added ablation study at the end of Table 2, it's interesting that removing demographic data doesn't decrease model performance despite age being consistently identified as the most important feature by the variable importance algorithms used in the papers. It would be helpful if the authors can add some thoughts or comments around this.

And it's helpful to see the model on all 314 LTLAs without conditioning on the dependent variable. To the concern that there may be co-linearity / curse of dimensionality effects when adding in the LTLA fixed effects, it might just be the case that hyperparameters (e.g., number of trees, max features, max depth, etc) need some tuning?

Response to editor and reviewers document

Manuscript: "Assessing the value of integrating national longitudinal shopping data into respiratory disease forecasting models"

Please see LaTeX manuscript for highlighted changes in blue. Please note websites and citation numbers also appear in blue.

Reviewer #2

Remarks to the Author:

1) Thanks to the authors for yet another round of thorough and detailed responses to previous comments. I appreciate the time and effort that went into addressing these concerns and am now in support of the paper for acceptance.

The additional text that has been added to qualify claims and state limitations or areas of possible concern is helpful.

1) Author response: Thank you, we appreciate these comments and are glad you found our additions helpful.

2) With regard to the added ablation study at the end of Table 2, it's interesting that removing demographic data doesn't decrease model performance despite age being consistently identified as the most important feature by the variable importance algorithms used in the papers. It would be helpful if the authors can add some thoughts or comments around this.

2) Author response: The demographic data did not include the variable 'age' but additional demographics. This was because age was done as a separate variable group and was included as part of the original baseline. To make sure this is now clear to the reader we have made this clear with added text in the table of the ablation study results (table 2 (second page), additional text: "(demographics variable group did not include age)" - noted the second half on table 2 has been moved to the supplement documents to meet Nature Communications formatting requirements and is now referred to as Supplement 5)

3) And it's helpful to see the model on all 314 LTLAs without conditioning on the dependent variable. To the concern that there may be co-linearity / curse of dimensionality effects when adding in the LTLA fixed effects, it might just be the case that hyperparameters (e.g., number of trees, max features, max depth, etc) need some tuning?

3) Author response: This indeed would be a concern, however, the model on all 314 LTLAs hyperparameters were optimized using the same technique as we used for all the models

created: a time series cross-validation grid-search to prevent overfitting. Therefore, optimized hyperparameters (number of trees, max features, max depth, etc) were specific to each model created and the variable inputs they included.